# Backdoor Secrets Unveiled: Identifying Backdoor Data with Optimized Scaled Prediction Consistency

**Soumyadeep Pal**[1], **Yuguang Yao**[2], **Ren Wang**[3], **Bingquan Shen**[4], **Sijia Liu**[2,5]
[1]University of Alberta, [2]Michigan State University, [3]Illinois Institute of Technology,
[4]DSO National Laboratories, [5]MIT-IBM Watson AI Lab, IBM Research

## Abstract

Modern machine learning (ML) systems demand substantial training data, often resorting to external sources. Nevertheless, this practice renders them vulnerable to backdoor poisoning attacks. Prior backdoor defense strategies have primarily focused on the identification of backdoored models or poisoned data characteristics, typically operating under the assumption of access to clean data. In this work, we delve into a relatively underexplored challenge: the automatic identification of backdoor data within a poisoned dataset, all under realistic conditions, *i.e.*, without the need for additional clean data or without manually defining a threshold for backdoor detection. We draw an inspiration from the scaled prediction consistency (SPC) technique, which exploits the prediction invariance of poisoned data to an input scaling factor. Based on this, we pose the backdoor data identification problem as a hierarchical data splitting optimization problem, leveraging a novel SPC-based loss function as the primary optimization objective. Our innovation unfolds in several key aspects. First, we revisit the vanilla SPC method, unveiling its limitations in addressing the proposed backdoor identification problem. Subsequently, we develop a bi-level optimization-based approach to precisely identify backdoor data by minimizing the advanced SPC loss. Finally, we demonstrate the efficacy of our proposal against a spectrum of backdoor attacks, encompassing basic label-corrupted attacks as well as more sophisticated clean-label attacks, evaluated across various benchmark datasets. Experiment results show that our approach often surpasses the performance of current baselines in identifying backdoor data points, resulting in about 4%-36% improvement in average AUROC. Codes are available at https://github.com/OPTML-Group/BackdoorMSPC.

## 1 Introduction

Deep neural networks (DNNs) have been a key component in driving a revolution in artificial intelligence over the past years, with applications in a variety of fields. They are used in computer vision (Krizhevsky et al., 2017; Goodfellow et al., 2020; Ren et al., 2015), autonomous driving (Wen and Jo, 2022), face detection (Yang et al., 2021) and various other realms. However, it has been shown in the literature that such deep networks are often brittle to various kinds of attacks. Attacks on DNNs can be test-time prediction-evasion attacks (Yuan et al., 2019), or they can occur through data poisoning attacks (Goldblum et al., 2022), which introduce feature and/or label pollution into the training data. In a recent survey conducted among industry professionals (Goldblum et al., 2022), it was discovered that the risks associated with data poisoning attacks are continually escalating for practitioners. In this work, we focus on a specific type of data poisoning attacks, called backdoor attacks. Backdoor attacks are usually performed by an adversary by polluting a small portion of the training dataset with (often) imperceptible backdoor triggers (*e.g.*, a small image patch on imagery data) such that it establishes a correlation with a target label. When models trained on such data are deployed, they can be easily manipulated by the application of such triggers in the test data.

Thus, an important avenue of research is developing backdoor defense methods against such attacks. Previous defense methods mostly focus on suppressing the backdoor attack (Du et al., 2019; Borgnia et al., 2021a;b; Pal et al., 2023; Hong et al., 2020; Liu et al., 2021a; Li et al., 2021a) or detecting

poisoned models (Chen et al., 2019; Kolouri et al., 2020; Wang et al., 2020; Shen et al., 2021; Xu et al., 2021). However, a more formidable challenge lies in directly identifying and pinpointing the backdoor samples concealed within the training set. On achieving such identification, users have much more freedom in their practice. In particular, a more intriguing inquiry pertains to how the aforementioned problem can be addressed under **practical conditions**. These include: **(P1) Free of Clean Data**: The user does not have access to *additional* clean base set prior to starting their training. **(P2) Free of Detection Threshold**: An identification algorithm may assign scores to training data samples which can reveal the likelihood of samples being backdoored, *e.g.*, a score by averaging training loss at early epochs (Li et al., 2021b). We present different scoring methods in Appendix A. In such a scenario, a user would need to manually set a detection threshold for these scores either heuristically or by knowledge of poisoning ratio (Zeng et al., 2022). We argue that in a realistic scenario, users should not be required to set such a manual threshold. We emphasize the importance of these practical conditions in Appendix B.

A class of defense methods like ABL (Li et al., 2021b), DBD (Huang et al., 2022), SD-FCT (Chen et al., 2022) roughly detect backdoor / clean samples, aimed at unlearning these samples for ease of downstream processing. However, it is worth noting that these methods are typically *not* designed for the accurate identification of backdoor samples, often leading to large errors in backdoor data selection. In **Table 1**, we denote the suitability of each existing method for 'Backdoor Data Identification' with a ✗ or ✓.

Several other methods are designed solely for backdoor identification. A

Table 1: **Practical conditions** for different backdoor identification methods. ✗ denotes undesired conditions and ✓ denotes desired conditions.

| Backdoor Defenses | Backdoor Data Identification | Practical Conditions | |
|---|---|---|---|
| | | **(P1)** Free of Clean Data | **(P2)** Free of Detection Threshold |
| ABL (Li et al., 2021b) | ✗ | ✓ | ✗ |
| DBD (Huang et al., 2022) | ✗ | ✓ | ✗ |
| SD-FCT (Chen et al., 2022) | ✗ | ✓ | ✗ |
| STRIP (Gao et al., 2019) | ✓ | ✗ | ✗ |
| AC (Chen et al., 2018) | ✓ | ✓ | ✗ |
| SS (Tran et al., 2018) | ✓ | ✓ | ✗ |
| SPECTRE (Hayase et al., 2021) | ✓ | ✓ | ✗ |
| SCAn (Tang et al., 2021) | ✓ | ✗ | ✓ |
| SPC (Guo et al., 2023) | ✓ | ✓ | ✗ |
| CD (Huang et al., 2023) | ✓ | ✗ | ✓ |
| Meta-SIFT (Zeng et al., 2022) | ✓ | ✓ | ✗ |
| ASSET (Pan et al., 2023) | ✓ | ✗ | ✓ |
| Ours | ✓ | ✓ | ✓ |

group of such methods (Chen et al., 2018; Tang et al., 2021; Hayase et al., 2021; Tran et al., 2018) use separable latent representations learnt by a classifier to distinguish between backdoor and clean data points. This strong *latent separability assumption* was challenged through adaptive attacks in (Qi et al., 2023), which mitigates this separation and subsequently, such identification methods. Moreover, we observe that they violate either **P1** or **P2** (**Table 1**-row 5-8). Recent backdoor identification methods like (Guo et al., 2023; Huang et al., 2023) rely on different backdoor signatures like the scaled prediction consistency and cognitive pattern. However, the former requires setting a manual detection threshold while the latter requires clean samples for accurate identification.

To the best of our knowledge, the most relevant work to ours is (Zeng et al., 2022), which aims to sift out clean samples from a backdoored dataset. The work develops a bi-level optimization formulation that aims to minimise the cross-entropy loss of clean samples while simultaneously maximise that for backdoor samples. However, we note that this selection of clean data *is a relatively easier problem* because of its abundance when compared to backdoor data. For example, let us consider a dataset with 45k clean samples and 5k backdoor samples. Precisely, the error rate of identifying 10% clean samples out of 45k will be much lower when compared to that of identifying 10% backdoor samples out of 5k; see more discussions in Section 5.2. Moreover, this method requires the prior knowledge of the poisoning ratio for accurately identifying all backdoor samples thus violating **P2**.

Given this, existing backdoor identification methods do not satisfy either **P1** or **P2** of the studied practical condition setting. We have demonstrated this comprehensively in Table 1 and Table A1. In our work, we leverage the scaled prediction consistency signature of backdoor data points to develop a novel backdoor identification method, which satisfies both of our practical conditions. **Our contributions** are summarized below:

❶ We peer into the usage of the scaled prediction consistency signature and provide various insights explaining its limitations.

❷ Propelled by these insights, we develop a novel loss function called Mask-Aware SPC (MSPC). Using this loss, we develop a practical algorithm to identify backdoor samples, satisfying both **P1** and

**P2**. Our algorithm treats the problem as a hierarchical data-splitting task and optimizes the MSPC loss function using bi-level optimization techniques.

❸ Lastly, we conduct a comprehensive evaluation of our method across a range of metrics, including AUROC and TPR/FPR, against a wide variety of backdoor attacks. This includes basic BadNets, CleanLabel attacks, and more sophisticated Warping-based backdoor attacks at various poisoning rates and across different datasets. Our approach often outperforms or performs at par with baseline methods (which do not satisfy the practical constraints).

## 2 RELATED WORK

**Backdoor attacks.** Backdoor attacks aim to inject the backdoor trigger to the target model, so that the target model will consistently misclassify the backdoor samples with the backdoor trigger to the target label and behave normally on the clean samples. Based on the knowledge of the adversaries, the backdoor attacks can be categorized into two tracks: data poisoning based attacks (Gu et al., 2017; Liu et al., 2018; Chen et al., 2017; Turner et al., 2019; Zhao et al., 2020; Nguyen and Tran, 2021; Li et al., 2021c; Taneja et al., 2022) and training manipulation based attacks (Garg et al., 2020; Lin et al., 2020; Shumailov et al., 2021; Bagdasaryan and Shmatikov, 2021; Tang et al., 2020; Doan et al., 2021). This paper will focus on the first category, the data poisoning based attacks. Attack methods such as BadNets (Gu et al., 2017), Trojan (Liu et al., 2018), and Blend (Chen et al., 2017) add simple trigger patterns like square patches or blend another figure into the background of backdoor samples, then mislabel them to the target class. The clean-label attack (Turner et al., 2019) is designed to poison only the target-class samples through adversarial attacks, so that the backdoor trigger injection avoids label manipulation and becomes more stealthy. Further, TUAP (Zhao et al., 2020) incorporates optimized universal adversarial perturbation into the clean-label attack and improves the attack success rate. Later, more invisible and sample-specific methods like WaNet (Nguyen and Tran, 2021) and ISSBA (Li et al., 2021c) are proposed to bypass the backdoor defenses that assume the backdoor trigger is sample-agnostic.

**Backdoor defenses beyond identification.** To defend against backdoor attacks, numerous methods have been developed and can be divided into different categories. Backdoor trigger recovery (Wang et al., 2019; Guo et al., 2019; Liu et al., 2019; Sun et al., 2020; Liu et al., 2022; Xiang et al., 2022; Hu et al., 2021) aims to synthesize the backdoor trigger used by the adversary while backdoor model reconstruction (Borgnia et al., 2021a; Huang et al., 2022; Li et al., 2021b; Pal et al., 2023) attempts to purify the backdoor model by eliminating the backdoor effect. Backdoor model detection (Chen et al., 2019; Kolouri et al., 2020; Wang et al., 2020; Shen et al., 2021; Xu et al., 2021) identifies whether a model is poisoned from training on backdoor samples. However, the focal point of our work is identifying backdoor samples from a given dataset. We have elucidated various backdoor identification methods in Table 1. As mentioned previously, these methods do not meet our *practical constraints*, which we will address in this paper.

## 3 PRELIMINARIES AND PROBLEM SETUP

**Backdoor attacks and defender's capabilities.** We consider the setting where the user receives a backdoor dataset from a third party or external source and can train their model using that dataset. Given a clean dataset $\mathcal{D} = \{(\mathbf{x}_i, y_i)\}_{i=1}^N = \mathcal{D}_m \bigcup \mathcal{D}_n$ and a poisoning ratio $\gamma = \frac{|\mathcal{D}_m|}{|\mathcal{D}|}$ (with $\gamma \ll 1$), the adversary injects the backdoor data $\mathcal{D}_b$ in place of $\mathcal{D}_m$. $\mathcal{D}_b$ is generated using a certain backdoor data generator $\mathcal{G}$ such that $\mathcal{D}_b = \{(\mathbf{x}', y_t) \,|\, \mathbf{x}' = \mathcal{G}(\mathbf{x}), (\mathbf{x}, y) \in \mathcal{D}_m\}$, where $y_t \neq y$ is a target label. Thus, the user receives the backdoor dataset $\mathcal{D}_p = \mathcal{D}_b \bigcup \mathcal{D}_n$, and training on this leads to a backdoor model vulnerable to test-time manipulation by the adversary.

Let $\mathcal{F}_{\boldsymbol{\theta}} : \mathcal{X} \to \mathbb{R}^C$ denote the ML model that we consider, parameterized by $\boldsymbol{\theta}$. Here $\mathcal{X}$ is the input space and the model gives as an output a real vector with $C$ dimensions (*i.e.*, $C$ classes). Hence, the predicted class of an input $\mathbf{x}$ by the model $\mathcal{F}_{\boldsymbol{\theta}}$ is given as $\arg\max_{[C]} \mathcal{F}_{\boldsymbol{\theta}}(\mathbf{x})$, where $[C] = \{1, 2, 3, ..., C\}$. In the rest of the paper, we use the shorthand of this notation as $\arg\max \mathcal{F}_{\boldsymbol{\theta}}(\mathbf{x})$.

In the above setting, the defender wants to identify backdoor samples from this dataset under the **practical conditions** as mentioned in Section 1. In this premise, the defender is *free to choose to train any model* on this dataset for backdoor identification.

**Warm-up: SPC alone is not sufficient.** Of the various backdoor signatures in the literature (see Section 1), we draw inspiration from the *scale invariance signature* of backdoor samples, *i.e.*, the SPC (scaled prediction consistency) loss (Guo et al., 2023). This is because: **(1)** Of the various signatures applicable for backdoor sifting, SPC satisfies the constraint of no prior clean data (**P1**); **(2)** SPC is computationally efficient; And **(3)** SPC does not rely on the latent separability assumption. Specifically, given an input datapoint $\mathbf{x}$ and a set of scales $\mathcal{S} = \{2, 3, ..., 12\}$, the vanilla SPC loss measures agreement in the prediction of $\mathbf{x}$ and that of scalar multiplication of $\mathbf{x}$:

$$\ell_{\text{SPC}}(\mathbf{x}) = \sum_{n \in \mathcal{S}} \frac{\mathbb{1}(\arg\max \mathcal{F}_{\boldsymbol{\theta}}(\mathbf{x}) = \arg\max \mathcal{F}_{\boldsymbol{\theta}}(n \cdot \mathbf{x}))}{|\mathcal{S}|}, \tag{1}$$

where $\mathbb{1}$ denotes the indicator function, $|\mathcal{S}|$ denotes the cardinality of the set of scales $\mathcal{S}$, and $n$ is a scaling constant. The work (Guo et al., 2023) showed that for backdoor samples, the predicted class of a particular sample remains the same even when multiplied by scales, due to the strong correlation established between the trigger and the target label. Hence, for such a backdoor sample and for a particular scale $n \in \mathcal{S}$, the indicator function in (1) gives a value close to $+1$, which is averaged over the number of scales.

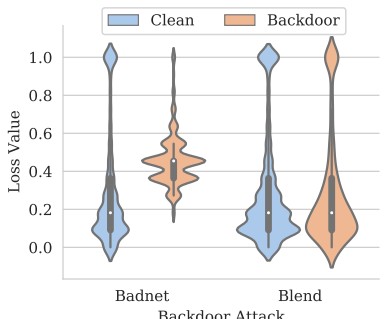

Figure 1: Violin plots for SPC loss for backdoor poisoned data and clean data when facing Badnet attack (Gu et al., 2017) and Blended attack (Chen et al., 2017).

Though the vanilla SPC loss has been shown to perform well in detecting backdoor samples over various attacks, we demonstrate some issues via **Figure 1**. **(1)** For Badnet attack (Gu et al., 2017), while the clean (or backdoor) samples have low (or high) SPC loss on average, we observe that there is a large variance in the loss values for both clean and backdoor samples. **(2)** The loss value distribution is indistinguishable for the Blended attack (Chen et al., 2017). **(3)** Moreover, to distinguish between clean and backdoor samples, the defender needs to set a threshold on SPC values (thus violating **P2**). Given the pros and cons of this signature, we ask:

> *(Q) Can we exploit the advantages of the scale invariance signature to develop a backdoor identification algorithm, which simultaneously satisfy P1 and P2 as introduced in Sec. 1?*

## 4 PROPOSED METHOD

**Exploring SPC limitations: Insights gained.** First, we delve into the vanilla SPC loss to gain insights from its various failure cases. From (1), backdoor samples are anticipated to exhibit high SPC loss values due to the scale-invariant nature of backdoors. However, instances where backdoor samples manifest low SPC loss values are explored, as illustrated in **Figure 2**. Two reasons for this anomaly are identified: Firstly, clipping pixel values to the range $[0, 1]$ after multiplying with a large scalar value causes the trigger to vanish (**Figure 2**-CIFAR10 row 1, 2 and ImageNet row 2); Secondly, the trigger may blend with the background at higher scales (**Figure 2**- ImageNet row 1). We have demonstrated the vanishing of the effective parts of the Blend trigger with CIFAR-10. As seen in **Figure 2**-CIFAR10 row 2, the trigger practically vanishes at $\times 7$, thus rendering the backdoor ineffective.

> **Insight 1.** *Backdoor samples* can obtain *low* SPC loss because the backdoor pixels vanish or blend with the background when multiplied with higher scales.

In contrast, we also examine instances of clean samples with high SPC losses, which deviate from expectations. This occurrence is prevalent when objects of interest maintain their structure despite scaling, primarily due to their low pixel values, as exemplified in **Figure 2**.

> **Insight 2.** *Clean samples* can obtain *high* SPC loss value because predictive features of the main object remain intact even at high scales.

Additionally, two less common scenarios are noted: Images completely vanish due to scaling, resulting in high SPC losses when the original class remains consistent (**Figure 2**-CIFAR10 row 4), and the emergence of certain spurious correlations when scaling, which maintain image predictions

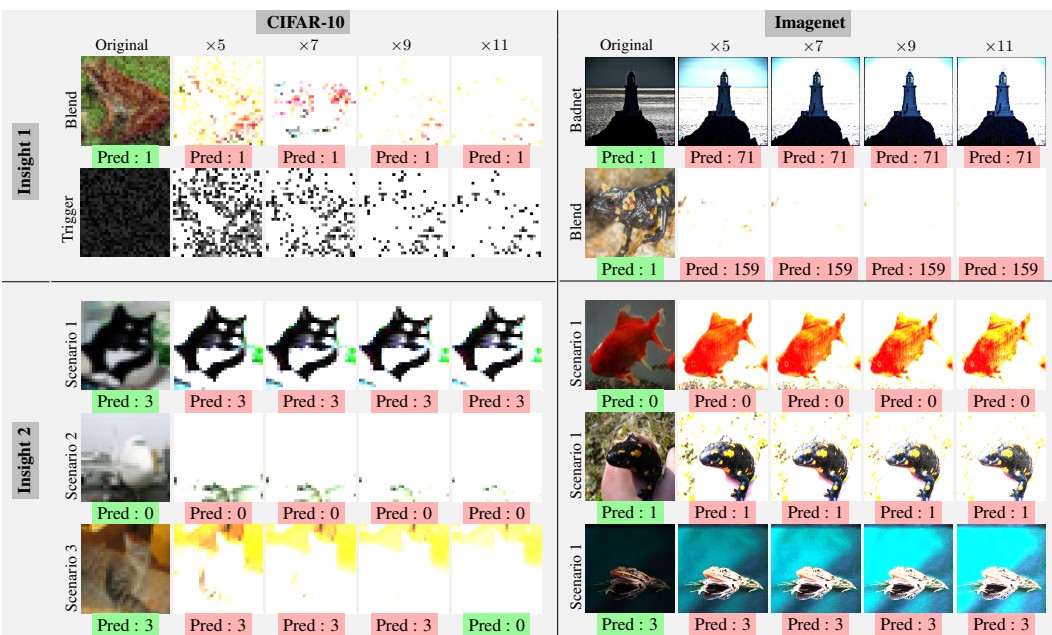

Figure 2: Illustration of the various SPC limitations in terms of the elucidated insights. ×a indicates an image is multiplied by scalar a and then constrained between 0 and 1. **Insight 1** : Backdoor samples with low SPC loss. **Insight 2** : Clean samples with low SPC loss. Predictions are marked as green if it is an expected prediction, while it is marked red for an unexpected prediction. Note that for backdoored samples, the expected label is 1 which is the target label.

(**Figure 2**-CIFAR10 row 5). This observation, detailed further in Appendix D, elucidates high SPC values for benign samples without systematically detecting spurious features.

**Enhancing SPC: Mask-aware SPC (MSPC).** Informed by the previous insights, we introduce a novel mask-aware SPC loss. Given a (known) mask $\mathbf{m}$ and a subtle linear shift $\tau$, we define the proposed MSPC loss below:

$$\ell_{\mathrm{MSPC}}(\mathbf{x}_i, \mathbf{m}, \tau) = \frac{1}{|S|} \sum_{n \in S} \phi(\arg\max \mathcal{F}_{\boldsymbol{\theta}}(\mathbf{x}_i) - \arg\max \mathcal{F}_{\boldsymbol{\theta}}(n \cdot \mathbf{x}_i^m)),$$

$$\mathbf{x}_i^m = (\mathbf{x}_i - \tau) \odot \mathbf{m}, \quad \phi(x) = \begin{cases} 1 & \text{if } x = 0 \\ -1 & \text{if } x \neq 0. \end{cases}$$

(2)

In (2), we consider a known mask $\mathbf{m}$ applied on an input $\mathbf{x}_i$, which is shifted by $\tau$ to form the masked image $\mathbf{x}_i^m = (\mathbf{x}_i - \tau) \odot \mathbf{m}$. The linear shift $\tau$ is to slightly shift the histogram of the trigger to lower pixel values. This can help in preserving the trigger at higher scales. We *assume* that the mask $\mathbf{m}$ has encoded the focused 'effective part' of the trigger in backdoor samples. This is reminiscent of (Huang et al., 2023), which proposed to extract the 'minimal essence' of an image for prediction, *i.e.*, the minimal mask such that the prediction of the model stays the same.

In such a scenario, $\ell_{\mathrm{MSPC}}$ preserves the core desirable behavior of $\ell_{SPC}(\mathbf{x})$: If $\mathbf{x}$ is a clean sample, the model prediction changes over scales and if $\mathbf{x}$ is a backdoor sample, the predictions remain consistent. We can see the advantage of using such a loss in **Figure 3**. *(1)* Row 1 visualises the mitigation of limitation of **Insight 1**. For backdoor images, the mask and linear shift are able to maintain a consistent (backdoor) target prediction. *(2)* Row 2 visualises the mitigation of limitation of **Insight 2**. Since the mask focuses on effective parts of the trigger, clean images are unable to maintain the same prediction at higher scales, even if object remains intact (Row 2 column ImageNet) or if there is background reliance (Row 2 column CIFAR-10). On applying the mask, since the network *does not* focus on the object, the prediction for clean images change with higher scales.

**Backdoor identification via bi-level optimization.** We propose a hierarchical data-splitting optimization method, serving as a fundamental element of our research, offering a solution to find $\mathbf{m}$ in

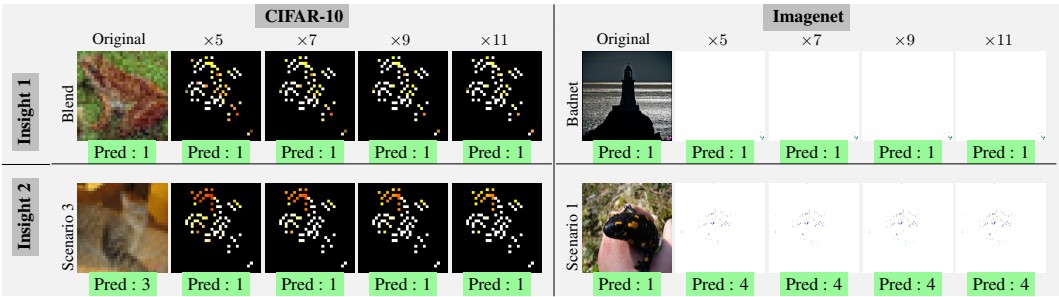

Figure 3: Advantage of using masks ($\mathbf{m}$ as described in 2). We note that here we use optimized masks obtained from our proposed method later. We apply a threshold of 0.008 for purposes of visualisation. For Imagenet, colors are inverted for ease of visualisation. Predictions are marked as green if it is an expected prediction, while it is marked red for an unexpected prediction. Note that for backdoored samples, the expected label is 1 which is the target label.

alignment with (2) and satisfying both constraints **P1** and **P2**. This approach enables us to maximize the potential of the proposed MSPC loss, incorporating inspiration from (Zeng et al., 2022).

Let $w_i \in \{0, 1\}$ (for $i \in [N]$) serve as the indicator of whether the training sample $\mathbf{x}_i$ is backdoored. If $w_i = 1/0$, then $\mathbf{x}_i$ is a backdoor/clean sample. First, we express the problem of *automatic* splitting of backdoor data as the following bi-level problem:

$$\min_{\mathbf{w} \in \{0,1\}^N} \quad \sum_{i=1}^{N}(1 - w_i)\, L(\mathbf{x}_i, \boldsymbol{\mu}^*(\mathbf{w})) \tag{3}$$
$$\text{subject to} \quad \boldsymbol{\mu}^*(\mathbf{w}) = \arg\max_{\boldsymbol{\mu}} \frac{1}{N} \sum_{i=1}^{N} w_i\, L(\mathbf{x}_i, \boldsymbol{\mu}).$$

Here $L$ is a loss function that needs to be designed such that it has a positive value for backdoor samples and a negative value for clean samples. Under such a scenario, the lower (upper) level is maximized (minimized) by the contribution of backdoor (clean) samples. In (3), the lower-level optimization variables are represented by $\boldsymbol{\mu}$, which can be utilized to characterize the masking variables $\mathbf{m}$ in MSPC.

Considering the scale invariance property of backdoor samples and leveraging the proposed use of masks $\mathbf{m}$, we propose the **lower-level optimization** of (3) to be:

$$\mathbf{m}^*(\mathbf{w}) = \arg\min_{\mathbf{m}} \frac{1}{N} \sum_{i=1}^{N} \Big[ w_i\, \frac{1}{|S|} \sum_{n \in S} D_{KL}(\mathcal{F}_{\boldsymbol{\theta}}(\mathbf{x}_i) || \mathcal{F}_{\boldsymbol{\theta}}(n \cdot \mathbf{x}_i^m)) \Big] + \lambda \|\mathbf{m}\|_1, \tag{4}$$

where recall that $\mathbf{x}_i^m = (\mathbf{x}_i - \tau) \odot \mathbf{m}$ and $D_{KL}(.||.)$ denotes the Kullback-Leibler (KL) divergence. We note that maximizing the KL divergence is akin to minimizing the SPC loss in (1). Moreover, since our mask $\mathbf{m}$ focuses on the 'effective part' of the trigger, we also penalize the size of the mask similar to (Huang et al., 2023). We refer readers to Appendix C for more details regarding the rationale behind the proposed lower-level optimization.

For the **upper-level optimization**, we propose the direct use of the MSPC loss. Thus, we finally achieve our proposed bi-level optimization problem:

$$\min_{\mathbf{w} \in \{0,1\}^N} \quad \sum_{i=1}^{N}(1 - w_i)\, \ell_{\text{MSPC}}(\mathbf{x}_i, \mathbf{m}^*(\mathbf{w}), \tau) \tag{5}$$
$$\text{subject to} \quad \mathbf{m}^*(\mathbf{w}) \text{ is obtained by (4)}.$$

To solve the above bi-level optimization problem, we relax $\mathbf{m}$ to lie between 0 and 1 by converting a combinatorial optimization problem to a continuous one. Then we solve the problem using simple alternating optimization, starting from the upper level (Liu et al., 2021b). We calculate the AUROC in Section 5.2 using this MSPC loss vector after optimization. Moreover, because of (2), we note backdoor samples will simply be the ones with MSPC loss greater than 0, thus satisfying **P2**. See Appendix C for further details on the rationale behind the proposed bi-level optimization.

## 5 EXPERIMENTS

### 5.1 EXPERIMENT SETUP

We evaluate our method on three different datasets including CIFAR-10 (Krizhevsky et al., 2009), Tiny-ImageNet (Deng et al., 2009) and ImageNet200 (Deng et al., 2009) (which is an ImageNet

Table 2: Mean AUROC values for our method and different baselines including SPC, ABL, SD_FCT, and STRIP on CIFAR10, Tiny Imagenet, and Imagenet200. The best result in each setting is marked in **bold** and the second best is underlined. All attacks have a poisoning ratio equal to 10% except the AdaptiveBlend attack. The last row shows the average performance of each method across all attacks.

| Attack | Methods | | | | |
|---|---|---|---|---|---|
| | **Ours** | SPC | ABL | SD-FCT | STRIP |
| **CIFAR10** | | | | | |
| Badnet | 0.9514 (0.0043) | 0.8318 (0.0566) | 0.8967 (0.0573) | **1.0000** (0.0000) | 0.9996 (0.0002) |
| Blend | 0.9526 (0.0012) | 0.5032 (0.0363) | 0.9335 (0.0329) | 0.9555 (0.0387) | **0.9956** (0.0033) |
| LabelConsistent | 0.9540 (0.0066) | 0.9246 (0.0012) | 0.6970 (0.0713) | 0.5418 (0.0267) | **0.9892** (0.0024) |
| TUAP | 0.9275 (0.0083) | 0.8128 (0.0104) | 0.7724 (0.0473) | 0.6920 (0.0270) | **0.9855** (0.0064) |
| Trojan | 0.9547 (0.0013) | 0.7688 (0.0120) | 0.9070 (0.0427) | 0.8605 (0.1074) | **0.9946** (0.0024) |
| Wanet | **0.9331** (0.0236) | 0.7259 (0.0210) | 0.8439 (0.0562) | 0.4120 (0.0038) | 0.8959 (0.0148) |
| DFST | 0.5750 (0.1748) | 0.3524 (0.0320) | 0.8475 (0.0410) | **0.8855** (0.0309) | 0.7676 (0.0454) |
| AdapBlend ($\gamma = 0.3\%$) | **0.9046** (0.0414) | 0.3313 (0.0128) | 0.4521 (0.0032) | 0.4226 (0.0072) | 0.1378 (0.0149) |
| Average | **0.8941** | 0.6563 | 0.7937 | 0.7212 | 0.8457 |
| **Tiny Imagenet** | | | | | |
| Badnet | **0.9983** (0.0004) | 0.9786 (0.0052) | 0.9559 (0.0252) | 0.9593 (0.0134) | 0.9968 (0.0009) |
| Blend | 0.9926 (0.0057) | 0.7120 (0.0093) | 0.9544 (0.0211) | 0.6863 (0.1375) | **0.9993** (0.0000) |
| Wanet | **0.9976** (0.0000) | 0.9914 (0.0017) | 0.9211 (0.0424) | 0.5709 (0.1071) | 0.9956 (0.0011) |
| Average | 0.9961 | 0.8940 | 0.9438 | 0.7388 | **0.9972** |
| **Imagenet200** | | | | | |
| Badnet | **0.9980** (0.0004) | 0.9655 (0.0042) | 0.9522 (0.0134) | 0.9053 (0.0142) | 0.8681 (0.0419) |
| Blend | 0.9836 (0.0015) | 0.6319 (0.0062) | 0.9352 (0.0369) | 0.4537 (0.0150) | **0.9916** (0.0024) |
| Average | **0.9908** | 0.7987 | 0.9437 | 0.6795 | 0.9298 |

subset of 200 classes) using ResNet-18 models (He et al., 2016). We provide training details for each of these datasets in Appendix E. We also provide some additional experiments using the Vision Transformer architecture (Touvron et al., 2021) in Appendix I. We note that we do not apply data augmentations during training because it can decrease the attack success rate (Liu et al., 2020). We also provide an ablation study of the hyperparameter $\tau$ in Appendix J.

**Attack baselines.** For our evaluation, we considered a variety of backdoor attacks: 1. blended attack (dubbed 'Blend') (Chen et al., 2017), 2. label consistent attack (dubbed 'LabelConsistent') (Turner et al., 2019), 3. clean label attack with universal adversarial perturbations (dubbed 'TUAP') (Zhao et al., 2020), 4. Trojan attack (Liu et al., 2018), and 5. warped network attack (dubbed 'Wanet') (Nguyen and Tran, 2021) 6. DFST (Cheng et al., 2021) 7. Adaptive-Blend (dubbed 'AdapBlend') (Qi et al., 2023). These included basic patch-based attacks, clean-label attacks, and more advanced non-patch-based attacks. We note that the Adaptive-Blend attack (Qi et al., 2023) violates the latent separability assumption, which is assumed by several backdoor identification methods. We evaluated the efficacy of our method at two different poisoning ratios ($\gamma$), 5% and 10%. Implementation details of our attack settings can be found in Appendix F.

**Evaluation.** We consider two quantitative evaluation metrics. (a) The Area under Receiver Operating Curve (AUROC): The ROC curve plots the True Positive Rate (TPR) against the False Positive Rate (FPR) of the algorithm for different thresholds of the loss value. AUROC measures the area under such a curve. A perfect detection method would have an AUROC of 1. **(b)** TPR and FPR of our method using our **algorithm's built-in threshold of 0**. We also compare our method (in terms of AUROC) with other baseline backdoor detection algorithms, *i.e.*, SPC (Guo et al., 2023), ABL (Li et al., 2021b), SD-FCT (Chen et al., 2022) and STRIP (Gao et al., 2019). We direct the readers to Appendix G for implementation details of the included baselines. Note that these baseline methods do not satisfy both **P1** and **P2** (Section 1). Unless explicitly stated otherwise, the reported results are averaged over 3 independent trials, with their means and variances provided. Standard deviations are presented in parentheses.

## 5.2 EXPERIMENT RESULTS

**High accuracy of backdoor identification.** First, we investigate the accuracy of our method in detecting backdoor samples. We present our key observations as follows: **(1)** In **Table 2**, we see that our detection algorithm achieves better AUROC than most baseline methods for attacks like LabelConsistent, TUAP, Trojan and Wanet for CIFAR-10. We note that our method is usually outperformed by STRIP, however STRIP requires a clean set to set their detection threshold, thus

violating **P2**. For ImageNet200, our method performs better than all other baselines. **(2)** Our method performs well against AdapBlend, an attack specifically designed against the latent separability assumption with a *poisoning rate of 0.003*. We see that all the baseline methods fail in this case. **(3)** Our method does not perform at par with other attacks when evaluated on DFST. This may be a potential limitation of our method. However, since we consider the problem of identification under **P1** and **P2**, we would like to emphasize our success in the rest of the diverse set of attacks on different datasets. **(4)** From **Figure 4**, we observe that our algorithm can achieve a TPR of 1.0 with a low FPR of 0.2, *i.e.*, it can identify *all* backdoor samples while only falsely predicting 20% of clean samples as backdoor.

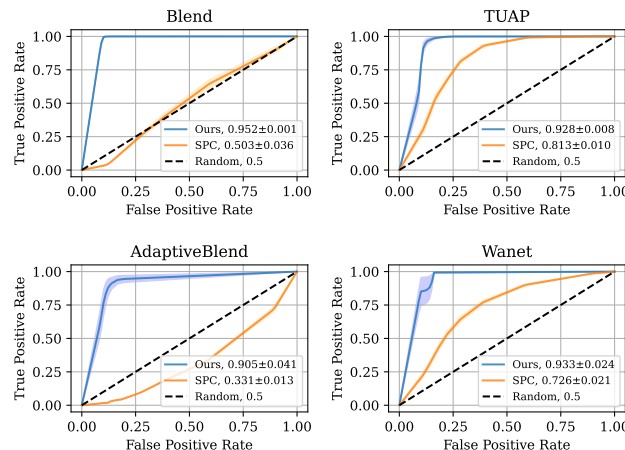

Figure 4: Selected AUROC plots for our method. Other baselines because they do not satisfy **P1** and **P2**, but SPC (Guo et al., 2023) is included for reference. Error bars indicate half a standard deviation over 3 runs.

**(5)** In **Table 3**, we demonstrate the effectiveness of the automated thresholding (at 0 MSPC loss). We draw attention to the fact that our method achieves very high TPRs, around 0.9 for most settings. Even with such high TPR rates, we achieve low FPRs in the range of 0.100-0.166 among all settings. In this context, we note that all other baselines would need manual thresholding (because of **P2** violation). We provide additional results (with $\gamma = 0.05$) in Appendix H.

Table 3: Average TPR/FPR values for our method and for reference SPC (Guo et al., 2023). Our algorithm has an **automated threshold of 0**. The threshold considered for SPC is 0.5. Results are presented in the format of 'TPR/FPR'.

| Attack | SPC | Ours |
|---|---|---|
| **CIFAR-10** | | |
| Badnet | 0.569 (0.296) / 0.189 (0.002) | 1.000 (0.000) / 0.156 (0.016) |
| Blend | 0.127 (0.007) / 0.183 (0.004) | 0.999 (0.001) / 0.104 (0.005) |
| LabelConsistent | 0.937 (0.004) / 0.178 (0.001) | 0.996 (0.006) / 0.143 (0.011) |
| TUAP | 0.588 (0.053) / 0.188 (0.003) | 0.990 (0.013) / 0.151 (0.024) |
| Trojan | 0.558 (0.026) / 0.181 (0.003) | 1.000 (0.000) / 0.102 (0.002) |
| Wanet | 0.439 (0.052) / 0.185 (0.001) | 0.879 (0.170) / 0.113 (0.019) |
| AdapBlend | 0.042 (0.006) / 0.182 (0.003) | 0.904 (0.087) / 0.12 (0.014) |
| **TinyImagenet** | | |
| Badnet | 0.955 (0.020) / 0.067 (0.002) | 1.000 (0.000) / 0.006 (0.001) |
| Blend | 0.315 (0.018) / 0.069 (0.001) | 0.979 (0.020) / 0.006 (0.001) |
| Wanet | 0.983 (0.010) / 0.038 (0.004) | 1.000 (0.000) / 0.005 (0.000) |
| **Imagenet200** | | |
| Badnet | 0.993 (0.010) / 0.157 (0.000) | 1.000 (0.000) / 0.006 (0.000) |
| Blend | 0.350 (0.011) / 0.153 (0.002) | 0.935 (0.007) / 0.003 (0.000) |

**Model retraining effect.** In **Table 4**, we present the results of retraining a model using only the clean samples identified by our method. We undertake model retraining to investigate the backdoor cleansing effects resulting exclusively from our algorithm. We observe that for most attacks, retraining the model can reduce the ASR to less than 0.52%, thereby rendering the backdoor ineffective. For the Wanet attack on CIFAR-10 and the Blend attack on TinyImageNet and ImageNet200, we note that model retraining can yield a high ASR. We emphasize that we *do not* advocate for retraining the model; instead, we aim to demonstrate the effectiveness of merely identifying the backdoor samples using our algorithm. As mentioned in Section 1, the user is free to take any action after identification (*e.g.*, unlearning).

Next, we demonstrate the difference between identifying clean samples and poison samples as mentioned in Section 1. We consider the problem of identifying a fraction of clean samples under the Badnet attack on CIFAR-10. In **Figure 5**, we observe that though Meta-SIFT (Zeng et al., 2022) is successful in accurate identification when sifting a low fraction of clean samples, but it degrades in performance when trying to sift out $80 - 100\%$ of clean samples. In sharp contrast, our method is successful in maintaining stable performance over all fractions of clean samples. No poison samples are present because of the TPR of 1 as presented in **Table 3**-row 1. We acknowledge that Meta-SIFT does not claim to identify backdoor samples with high accuracy. Yet, Figure 5 emphasizes that accurately finding poison samples is harder.

Table 4: Effect of retraining models without backdoor samples identified by our algorithm. ACC denotes standard accuracy, ASR denotes the Attack Success Rate.

| Attack | Before | | After | |
|---|---|---|---|---|
| | ACC | ASR | ACC | ASR |
| **CIFAR10** | | | | |
| Badnet | 88.36 (0.12) | 100.00 (0.00) | 81.35 (0.89) | 0.52 (0.60) |
| Blend | 88.29 (0.32) | 100.00 (0.00) | 79.28 (0.89) | 3.85 (5.45) |
| LabelConsistent | 89.36 (0.42) | 99.31 (0.33) | 83.08 (1.38) | 2.59 (3.62) |
| Trojan | 88.62 (0.09) | 100.00 (0.00) | 79.58 (0.71) | 0.00 (0.00) |
| Wanet | 88.20 (0.20) | 98.53 (0.18) | 79.40 (1.06) | 32.94 (46.59) |
| **Tiny Imagenet** | | | | |
| Badnet | 53.04 (0.53) | 99.24 (0.20) | 54.09 (0.20) | 0.14 (0.06) |
| Blend | 53.16 (0.26) | 99.84 (0.04) | 53.55 (0.26) | 32.41 (32.06) |
| **Imagenet200** | | | | |
| Badnet | 62.43 (0.29) | 99.99 (0.00) | 62.39 (0.18) | 0.08 (0.03) |
| Blend | 62.04 (0.18) | 99.61 (0.04) | 62.61 (0.13) | 89.29 (1.58) |

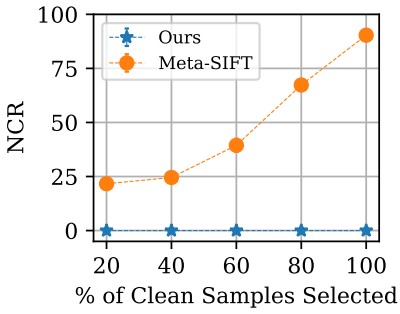

Figure 5: Normalized Corruption Ratio for different percentages of clean samples sifted out. If **x** clean samples are identified, NCR = ((number of poison samples in **x**) / **x**) / poison ratio $\times 100\%$.

**Robustness against potential adaptive attacks.** In what follows, we evaluate our method against potential white-box adaptive attacks, *i.e.*, the adversary has full knowledge about our detection method. We note that the adversary cannot manipulate the training process, but can change the input to evade our proposed detection. We consider the possibly strongest adversary that has access to an optimized mask $\mathbf{m}^*(\mathbf{w})$ from the upper level of our bi-level formulation in (5). Given this mask, the adversary can find a trigger $\mathbf{t}$ that aims to reduce the MSPC loss for poisoned samples thus acting in direct opposition to our algorithm. Specifically, given a poisoned model $\mathcal{F}_{\boldsymbol{\theta}}$, the adversary solves the following task :

$$\mathbf{t}^* = \arg\max_{\mathbf{t}} \ \frac{1}{N} \sum_{i=1}^{N} \Big[ \frac{1}{|S|} \sum_{n \in S} D_{KL}(\mathcal{F}_{\boldsymbol{\theta}}(\mathbf{x}_i) || \mathcal{F}_{\boldsymbol{\theta}}(n \cdot \mathbf{x}_i^m)) \Big] \tag{6}$$

We consider a Badnet-like scenario with an initial MSPC loss of 0.9910 (MSPC of clean samples = -0.6837). Upon completing the above optimization, the MSPC for backdoor samples drops to -0.7248, while maintaining a 100% ASR; thus, the above optimization is, in fact, successful. Using this trigger with a poison ratio of 0.1, we train a new model to evaluate our method. Our method achieves an AUROC of 0.9546 with a standard deviation of 0.3333 over three runs, indicating that the adaptive attack (which directly aims to mitigate our methodology) fails to reduce its effectiveness. We note that a possible reason for this attack's failure is that the optimized mask may not be unique or that the effectiveness of such an adaptive trigger is not training-agnostic.

## 6  CONCLUSION

In this paper, we address the lesser-explored challenge of automatically identifying backdoor data in poisoned datasets under realistic conditions, without requiring additional clean data or predefined thresholds for detection. We approach backdoor data identification as a hierarchical data splitting optimization problem, employing a novel Scaled Prediction Consistency (SPC)-based loss function. We refine the SPC method, develop a bi-level optimization-based approach to identify backdoor data precisely, and demonstrate our method's effectiveness against various backdoor attacks on different datasets. Our method does not perform at par with other baselines when evaluated on DFST. This may be a potential limitation of our method. However, since we consider the problem of identification under minimal assumptions (contrary to previous methods), we would like to emphasize our success in the rest of the diverse set of attacks on different datasets. We encourage future research on identifying backdoor samples, with the realistic constraints to focus on such deep feature space attacks and hope that our work lays a solid premise to do so.

## 7  ACKNOWLEDGEMENT

We extend our gratitude to the DSO National Laboratories for their support of this project. The contributions of Y. Yao and S. Liu are also partially supported by the National Science Foundation

(NSF) Robust Intelligence (RI) Core Program Award IIS-2207052. R. Wang is supported by the NSF under Grant 2246157 and the ORAU Ralph E. Powe Junior Faculty Enhancement Award.

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

APPENDIX

# A  PRACTICAL CONDITIONS OF VARIOUS DEFENSE METHODS

Table A1: Practical conditions, different scoring and thresholding mechanisms used by different backdoor identification methods. $\gamma$ indicates the actual poisoning ratio. $\epsilon$ indicates an upper bound on the poisoning ratio.

| Method | Realistic Conditions | | Identification Scores | Thresholding Method |
|--------|----------------------|--------------------------|-----------------------|---------------------|
| | **(P1)** Free of Clean Data | **(P2)** Free of Detection Threshold | | |
| ABL (Li et al., 2021b) | ✓ | ✗ | Training Loss | Heuristic |
| DBD (Huang et al., 2022) | ✓ | ✗ | Symmetric Cross Entropy Loss | Heuristic |
| SD-FCT (Chen et al., 2022) | ✓ | ✗ | FCT Metric | Heuristic |
| STRIP (Gao et al., 2019) | ✗ | ✗ | Entropy | Estimated from Clean Data |
| AC (Chen et al., 2018) | ✓ | ✗ | Silhoutte Scores of Activation Clusters | Heuristic |
| SS (Tran et al., 2018) | ✓ | ✗ | Inner Product between sample representation and top singular vector of feature representation matrix | $1.5\epsilon$ |
| SPECTRE (Hayase et al., 2021) | ✓ | ✗ | QUEScore | $1.5\epsilon$ |
| SCAn (Tang et al., 2021) | ✗ | ✓ | - | - |
| SPC (Guo et al., 2023) | ✓ | ✗ | SPC Loss | Heuristic |
| CD (Huang et al., 2023) | ✗ | ✓ | Norm of optimized "minimal" mask | Estimated from Clean Data |
| Meta-SIFT (Zeng et al., 2022) | ✓ | ✗ | Weights assigned by a Meta Network | Manual based on $\gamma$ |
| ASSET (Pan et al., 2023) | ✗ | ✓ | Optimized Loss | Automated |
| Ours | ✓ | ✓ | MSPC Loss | **Automated (at 0 loss)** |

In this section, we discuss the scores and the thresholding method used by relevant backdoor identification methods - which determines their satisfiability of the practical conditions defined in Section 1.

Anti Backdoor Learning (ABL) (Li et al., 2021b) consists of two steps - backdoor isolation and backdoor unlearning. In backdoor isolation, a local gradient ascent strategy is employed to amplify the difference in cross-entropy loss between the backdoor and benign samples. Based on a heuristic threshold on the training losses, a percentage of samples with low training losses are considered backdoor and further used for unlearning the backdoor samples.

Backdoor Defense via Decoupling (DBD) (Huang et al., 2022) first learns a purified feature extractor using semi-supervised learning and then further learns a classification network on top of this extractor. This is done by the SCE (Symmetric Cross Entropy) loss, where the clean samples obtain a low loss on average than the backdoor samples. A heuristic threshold can be used to distinguish between potential backdoor and clean samples - e.g. $50\%$ of the samples with the lowest loss are considered clean samples.

The sample distinguishment (SD) module from (Chen et al., 2022) uses the FCT metric to distinguish between backdoor and clean samples, however these are used downstream for backdoor removal via unlearning or secure training via a modified contrastive learning. The FCT metric measures the distance between network features for a sample and its simple trasformation (like rotation etc.). Chen et al. (2022) sets two thresholds that divides the FCT score into a poisoned set, a clean set and an uncertain set.

Thus all of the above methods violate **P2** because of heuristic thresholds.

STRIP (Gao et al., 2019) finds the average entropy of a given sample over various perturbations (by superimposition with a base set of images). A low entropy (i.e. low randomness for change in prediction upon superposition) indicates a backdoor sample. However, to identify backdoor samples, a detection boundary needs to be set which is typically done by estimating the entropy distribution statistics of clean inputs, thus violating **P1** and **P2**.

Spectral Signature (SS) (Tran et al., 2018) demonstrates the separability of poisoned and clean representations using correlations with the top singular vector of the representation matrix of samples. Thus, they compute an outlier score based on this to detect backdoor samples. Considering $\epsilon$ to be an upper bound on the number of backdoor samples, samples with the top $1.5\epsilon$ scores are considered backdoor. Since setting $\epsilon$ is heuristic (and requires an approximate knowledge of the poisoning ratio), this violates **P2**.

SPECTRE (Hayase et al., 2021) uses similar statistical properties of representations (whitened using robust estimation of mean and standard deviation of clean data) called QUantum Entropy Score

(QUEscore). However, similar to (Tran et al., 2018), the top $1.5\epsilon$ scores are considered to be backdoor samples thus violating **P2**.

We note that SCAn (Tang et al., 2021), does not use such identification scores - rather it relies on hypothesis testing to test whether representations of samples from a specific class follows a multivariate mixture distribution (denoting backdoor) or a single distribution (denoting benign). However, it uses clean data for the hypothesis testing and violates **P1**.

Scaled Prediction Consistency (SPC) (Guo et al., 2023) uses the scale-invariant backdoor signature (which we describe before). However, the detection threshold of the SPC loss (samples with loss above this threshold is considered backdoor) is either determined heuristically or through an additional set of clean data. In this paper, we consider the milder condition of unavailability of clean data - thus only using SPC violates **P2**.

Cognitive Distillation (CD) (Huang et al., 2023) uses the cognitive pattern signature. Cognitive pattern refers to the minimal distilled pattern (given by a mask) required for a particular sample to retain its original prediction from a trained model. The work showed that the $l_1$ norm of the CP of backdoor samples tend to be much smaller than that of clean samples. However, to set the threshold over the norms, the presence of a clean set is assumed, thus violating **P1**.

In Meta-SIFT (Zeng et al., 2022), a bi-level optimization algorithm is formulated that separates the clean and backdoor samples in the upper and lower level of their formulation using weights , similar to (3) . However, the weights are given by a meta-network - based on which a fraction of clean samples are selected. Thus, in this scenario, to identify backdoor samples accurately, ideally one should know the poison ratio or they need to set a heuristic threshold (violating **P2**).

## B  IMPORTANCE OF THE PRACTICAL CONDITIONS

In this section, we emphasize the importance of the practical conditions that we introduce in our paper.

**Importance of P1 (Free of Clean Data)**:

We provide several examples to understand the importance of the clean data assumption:

**(1)** Firstly, we provide an example where it may be very hard / impossible to obtain clean data. We consider the case of prediction of the presence of Autism Spectrum Disorder (ASD) in children. Recent studies (Tariq et al., 2018) collect home videos of children (submitted by parents) via crowdsourcing. Parents are required to take home videos of their child and upload them to a portal. Such crowdsourcing methods maybe highly susceptible to backdoor attacks.

In such cases, it may be very hard to collect clean data because that would require monitoring of the video capturing and uploading. This can be difficult or impossible because of various reasons (eg. change in behaviour of children, inavailability of consent to monitoring etc.)

**(2)** In some cases, organizations (which are clients to various ML companies) may be reluctant to provide / collect clean data for their application. For example, in a recent industry survey (Grosse et al., 2023), companies included backdoor attacks as one of their security concerns. Such organizations tend to push the security responsibility upstream to service providers as mentioned in (Grosse et al., 2023) by one of the respondents - "We use Machine Learning as a Service and expect them to provide these robust algorithms and platforms".

We believe many such scenarios associated with impossibility or reluctance of clean data collection may arise and are only limited to our imagination. This necessitates the development of algorithms without clean data assumption.

**Importance of P2 (Free of Detection Threshold)**:

We consider the case where this assumption is violated. Under such a condition, identification of backdoor datapoints requires the user to set some threshold on the scores assigned to each training data sample. Such a threshold depends on various factors like the poisoning ratio, the backdoor attack type and the dataset being used. For example, defense methods like SPECTRE (Hayase et al., 2021) and Spectral Signature (Tran et al., 2018) assume the knowledge of some upper bound of the poisoning ratio. On the other hand, CD (Huang et al., 2023) suggests calculating the threshold

from the mean and standard deviation of a subset of clean samples. We provide the various existing thresholding methods in **Table A1** in the Appendix A.

Thus, without some form of prior knowledge like the poisoning ratio (which is seldom the case), it is almost impossible for an user to determine such a threshold value. Thus, this indicates the importance of our assumption P2.

## C   DETAILS FOR THE BI-LEVEL OPTIMIZATION FORMULATION

In this section, we will describe the strategies we adopt to develop the bi-level formulation (Zhang et al., 2023) (5). Recall the data splitting problem of (3) :

$$
\begin{aligned}
&\min_{\mathbf{w} \in \{0,1\}^N} && \sum_{i=1}^N (1 - w_i)\, L(\mathbf{x}_i, \boldsymbol{\mu}^*(\mathbf{w})) \\
&\text{subject to} && \boldsymbol{\mu}^*(\mathbf{w}) = \arg\max_{\boldsymbol{\mu}} \ \frac{1}{N} \sum_{i=1}^N w_i\, L(\mathbf{x}_i, \boldsymbol{\mu})
\end{aligned}
\tag{A1}
$$

We consider masking variable $\mathbf{m}$ in MSPC as the optimization variable $\boldsymbol{\mu}$ in (3). In addition, the original SPC optimization is considered for an *upper-level* problem that involves the lower-level solution $\boldsymbol{\mu}^*(\mathbf{w})$, which is also a function of the upper-level variables $\mathbf{w}$.

Given the generic bi-level form in (3), both the upper and lower level optimizations work cohesively towards distinguishing between the backdoor samples and the clean ones. The lower level optimization tackles this challenge by amplifying the representation of backdoor samples within the loss. Conversely, the upper level optimization seeks to achieve the same goal by reducing the contribution of the poisoned samples within $L$. We note that this framework is different from bi-level optimization used for sifting out clean samples in Zeng et al. (2022), where $L$ is simply the cross entropy loss and $\boldsymbol{\mu}$ represents the parameters of a neural network model. We next need to integrate the proposed MSPC loss with the bi-level framework (3).

However, the MSPC loss is non-differentiable because of the occurrence of the function $\phi(x)$ in (2). We will now describe the strategies we adopt in each optimization level, to mitigate such issues, leading to our proposed bi-level solution.

First, we note that the basis of scaled prediction consistency is that for poisoned samples, the predicted class of a model for an input $\mathbf{x}$ remains consistent even when multiplied with scales, *i.e.*, $\arg\max \mathcal{F}_{\boldsymbol{\theta}}(\cdot)$ remains the same. Thus, for a sample $\mathbf{x}_i$ and for a scale $n \in \mathcal{S}$, maximizing the SPC loss would result in maximizing a non-differentiable loss $\mathbb{1}(\arg\max \mathcal{F}_{\boldsymbol{\theta}}(\mathbf{x}) = \arg\max \mathcal{F}_{\boldsymbol{\theta}}(n \cdot \mathbf{x}))$ for SPC or maximizing the non-differentiable $\phi(\arg\max \mathcal{F}_{\boldsymbol{\theta}}(\mathbf{x}) - \arg\max \mathcal{F}_{\boldsymbol{\theta}}(n \cdot \mathbf{x}))$ for the MSPC loss. In such a scenario, we argue that the probability distribution that we get from the model for poisoned inputs across scales would also remain similar, *i.e.*, the distance between $\mathcal{F}_{\boldsymbol{\theta}}(\mathbf{x})$ and $\mathcal{F}_{\boldsymbol{\theta}}(n \cdot \mathbf{x})$ would remain small when $\mathbf{x} \in \mathcal{D}_b$. Based on that, we can simply replace the non-differentiable loss maximization with the minimization of the Kullback-Leibler (KL) divergence. Specifically, this involves minimizing the divergence between the probability distribution output for a sample $\mathbf{x}_i$ and the output for the same sample when masked with $\mathbf{m}$ and subsequently scaled with $n$. This is in line with the proposed use of masks $\mathbf{m}$. In mathematical terms, this is denoted as $\min_{\mathbf{m}} D_{KL}(\mathcal{F}_{\boldsymbol{\theta}}(\mathbf{x}_i) || \mathcal{F}_{\boldsymbol{\theta}}(n \cdot \mathbf{x}_i^{\mathbf{m}}))$.

Leveraging the advantages of the proposed linear shift by $\tau$, we can customize the **lower-level optimization** of (3) to

$$
\mathbf{m}^*(\mathbf{w}) = \arg\min_{\mathbf{m}} \ \frac{1}{N} \sum_{i=1}^N \left[ w_i \ \frac{1}{|S|} \sum_{n \in S} D_{KL}(\mathcal{F}_{\boldsymbol{\theta}}(\mathbf{x}_i) || \mathcal{F}_{\boldsymbol{\theta}}(n \cdot \mathbf{x}_i^m)) \right] + \lambda \|\mathbf{m}\|_1,
\tag{A2}
$$

where recall that $\mathbf{x}_i^m = (\mathbf{x}_i - \tau) \odot \mathbf{m}$. The above addresses the challenge of optimizing the mask variable in a non-differentiable loss function by converting the lower-level optimization to a differentiable one, achieved through utilizing KL divergence.

As mentioned in Section 4, we simply use the MSPC loss for the **upper-level optimization**, thus achieving (5).

We note that the upper level is intricately linked with the lower level optimization, rendering the problem more complex. In practical terms, we managed to simplify this by treating the $\mathbf{m}^*(\mathbf{w})$ term

in the upper optimization as a constant value, considering only the $\mathbf{w}$ variable outside the MSPC loss. In this way, we can still use the MSPC loss in the upper-level optimization.

To solve the problem, we initialize the mask with the value of 1 for all pixels. We note that minimizing the upper level optimization is trivial in our case: we set $w_i = 1$ if $\ell_{\mathrm{MSPC}}(\mathbf{x}_i, \mathbf{m}, \tau)$ is positive and $w_i = 0$ if $\ell_{\mathrm{MSPC}}(\mathbf{x}_i, \mathbf{m}, \tau)$ is negative. We choose an odd number of scales for our experiments so that the value of the MSPC loss is never 0. During this upper level optimization, we consider the mask $\mathbf{m}(\mathbf{w})$ to be fixed. Solving the upper level optimization yields us a $\mathbf{w}$, which we use to solve the lower level optimization using gradient descent. We can perform this process iteratively to find the final MSPC loss vector where the $i$th value of this vector is given by $\ell_{\mathrm{MSPC}}(\mathbf{x}_i, \mathbf{m}^*(\mathbf{w}^*), \tau)$. This approach of alternating optimization, although a compromise, yielded satisfactory results (as demonstrated in Section 5.2) and significantly reduced the complexity of our optimization process.

## D  UNDERSTANDING SCENARIO 3 IN SECTION 4

We revisit the phenomenon of the Scenario 3 (**Figure 2**-CIFAR10 row 5) that we described in Section 4. We would like to provide a more detailed explanation of why the observed phenomenon reveals a spurious correlation.

Neural networks often rely on spurious features for their prediction (Neuhaus et al., 2022). The problem of spurious correlation can come from strong association between labels and backgrounds in classification (Sagawa et al., 2020) (*e.g.*, Waterbird or CelebA dataset) (Sagawa et al., 2019), or by relying on artifacts (Bissoto et al., 2020) or simply by background reliance (Moayeri et al., 2022). We discover an intriguing phenomenon similar to the effect of spurious correlation: Certain spurious image patterns can be found simply when scaling the images by scalars. The predictions of images remain intact because of their reliance on such spurious features, as shown in **Figure 2**-column CIFAR-10 row 5. We emphasize that this is not a systematic detection of spurious features, rather an observation which explains high SPC values for benign samples.

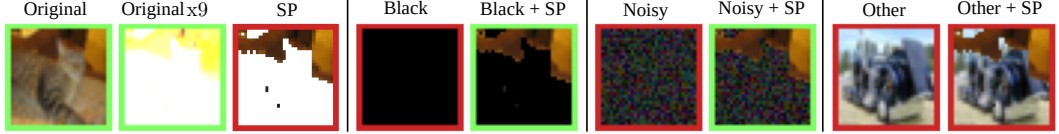

Figure A1: Figure indicates the presence of Spurious Correlation. 'x+ SP' indicates an image formed by stamping the Spurious Pattern (SP) into $\mathbf{x}$. Here SP is obtained by masking out the pixels of the original 'cat' image which stay (*i.e.* pixel values ¡ 1 across all 3 channels), when multiplied with the scaling factor 9. We display SP with a white background. 'Black' denotes a completely black image (with zero pixel values). 'Noisy' denotes an image drawn from a uniform distribution between 0 and 0.3. 'Other' indicates another image from the dataset CIFAR-10. The green box indicates that the predicted label is the same as 'cat' and the red box indicates that the predicted label is different from 'cat'.

In **Figure 2**-row 5, we observed that the main features of the original class (*i.e.* the cat) vanished when multiplied with higher scalar values. However, the model continued predicting the image as a cat. We notice that a yellow patch in that Figure remained. This naturally leads to the question of *whether that remaining patch causes the model to classify the image as cat*. To investigate this, we track the model's prediction when this patch is stamped onto various backgrounds. As seen in **Figure A1**, when this patch is pasted in a black or slightly noisy image, the model still predicts it as the class of cat, in spite of the fact that there exist no predictive features of a cat. This indicates the presence of a spurious correlation. However, as seen in **Figure A1**-column 4, the spurious correlation is not strong enough to affect the model prediction in the presence of other objects.

## E  TRAINING DETAILS.

We notice that $\tau = 0.1$ is sufficient to ensure the success of our method, where $\tau$ is the small linear shift of a sample $\mathbf{x}$ as proposed in (2). As mentioned in Section 4, We choose a series of scaling factors $\{2, 3, ..., 12\}$ following (Guo et al., 2023) in our experiments. For CIFAR-10 experiments, we solve our bi-level optimization problem using 10 epochs of minibatch SGD with a batch size

Table A2: Details for training models with different datasets used in this paper

| Dataset | Epochs | Initial Learning Rate | Learning Rate Scheduler | Learning Rate Decay Factor | Learning Rate Decay Epoch | Momentum | Weight Decay |
|---|---|---|---|---|---|---|---|
| CIFAR-10 | 182 | 0.1 | MultiStep LR | 0.1 | 91,136 | 0.9 | 5e-4 |
| TinyImageNet | 100 | 0.1 | MultiStep LR | 0.1 | 40,80 | 0.9 | 5e-4 |
| ImageNet200 | 60 | 0.5 | Cyclic LR | - | - | 0.9 | 5e-4 |

of 1000 and a learning rate of 0.1 for the lower level, with a momentum of 0.9 and a weight decay ratio of $5 \times 10^{-4}$. We perform the optimization for a total of 4 outer epochs. For TinyImageNet and ImageNet200 experiments, we perform only 5 epochs for the lower level. All experiments are conducted using 4 NVIDIA GeForce RTX 3090 GPUs with Pytorch implementation. We used FFCV (see https://ffcv.io/) to speed up our training process.

## F  BACKDOOR ATTACK SETTING DETAILS

We consider 6 widely used backdoor attacks for our experiments in this paper. We describe the detailed settings of each of those attacks as follows.

- Badnet (Gu et al., 2017) : We consider a $5 \times 5$ RGB trigger at the bottom right corner of our images.

- Blend (Chen et al., 2017) : As reported in the literature (Chen et al., 2017), we use random patterns as triggers and blend them in the image. Specifically, for an image $\mathbf{x}$ and the trigger pattern $\mathbf{t}$, the backdoor sample $\mathbf{x}' = 0.8\mathbf{x} + 0.2\mathbf{t}$.

- LabelConsistent (Turner et al., 2019) : This attack is performed by adding adversarial perturbations, generated using projected gradient descent bounded by $\ell_\infty$ norm, to images from the target class. We directly follow the above generation process detailed in (Huang et al., 2023). Moreover, the backdoored images are also stamped with $3 \times 3$ trigger (pixel values 0 or 255) at the bottom right corner.

- TUAP (Zhao et al., 2020): This attack involves a universal adversarial perturbation of images from the target class. This was performed using a 10-step projected gradient descent $\ell_\infty$ attack on a clean trained ResNet-18 model with maximum perturbation of $\epsilon = 8/255$ and a learning rate of $2/255$. Additionally, a Badnet-like trigger was attached to those images for the backdoor attack.

- Trojan (Liu et al., 2018) : We use a trigger given by Li et al. (2021a), which was constructed by reverse-engineering the last fully-connected layer of the network as proposed in the original paper (Liu et al., 2018).

- Wanet (Nguyen and Tran, 2021) : We follow the framework of (Nguyen and Tran, 2021) to generate a grid that is used to warp images to form the corresponding backdoor samples, apart for a few changes to ensure successful poisoning in our framework. Specifically, we start with an uniform grid of size $32 \times 32$ (similar to Huang et al. (2023)) with random values between $-1$ and $+1$. We choose the warping strength (Nguyen and Tran, 2021) to be 0.5, which is consistent with the setting in the original paper. This grid acts as the flow field, used to warp images. Similar to Huang et al. (2023), we also do not manipulate the training objective, thus staying consistent with our framework.

- DFST (Cheng et al., 2021) : This involves a style transfer of the input image using a cycleGAN. The style transfer is done using a 'sunrise weather' style used by Cheng et al. (2021). We use the poisoned data provided by Huang et al. (2023).

- AdapBlend (Qi et al., 2023) : The Adaptive Blend attack is performed similar to the Blend attack using the 'hellokity' blending pattern with modifications to reduce the latent separability between clean and backdoor samples. Specifically, the trigger is partitioned into 16 pieces - of which $50\%$ are used at train time and all are used at test time. Secondly, among the samples that are blended with the trigger, the labels of $50\%$ of them are replaced with the target label , while the others preserve the actual label (conservatism ratio = 0.5). Finally, an extremely low poisoning ratio of 0.003 is considered for this attack.

We use the target label of class 1 for the label-corrupted attacks in our experiments. All models trained using the backdoor datasets can achieve an acceptable clean accuracy and can be successfully attacked.

## G  BASELINE DEFENSE CONFIGURATIONS.

We describe the detailed settings of the baselines considered in this paper as follows:

- SPC (Guo et al., 2023) : We consider the scaling factors $\{2, 3, ..., 12\}$ to compute the SPC loss and use a threshold of 0.5 to find the TPR / FPR values

- ABL (Li et al., 2021b) : We consider the flooding loss (to avoid overfitting as described in (Li et al., 2021b)) in the gradient ascent step with a flooding parameter of 0.5. Then we consider 10 % of samples with the lowest losses over 10 epochs as backdoor samples.

- SD-FCT (Chen et al., 2022) : The Feature Consistency metric is computed using a transformation set consisting of random 180 degrees rotation and random affine transformations.

- STRIP (Gao et al., 2019) : To calculate the entropy, we took 100 randomly selected clean images from the validation set and used them to blend in the samples with equal weightage to both.

# H  ADDITIONAL RESULTS

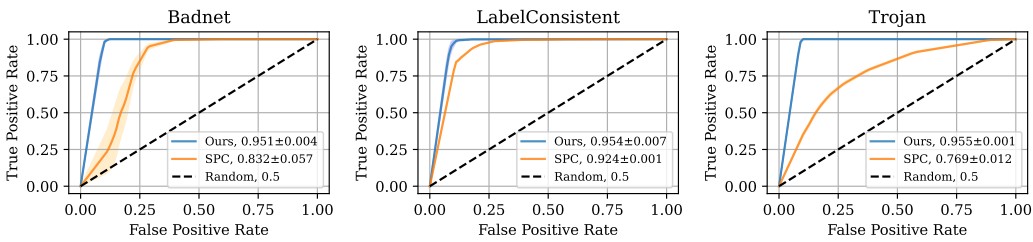

Figure A2: Additional ROC plots for our method with reference SPC (Guo et al., 2023) using CIFAR-10. Error bars indicate half a standard deviation over 3 runs. The poisoning ratio is 0.1.

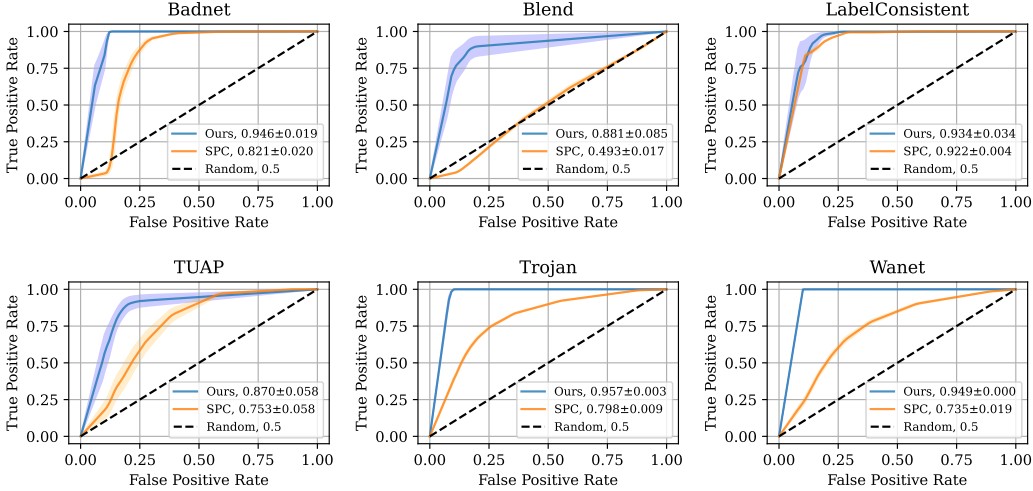

Figure A3: ROC plots for our method with reference SPC (Guo et al., 2023) using CIFAR-10. Error bars indicate half a standard deviation over 3 runs. The poisoning ratio is 0.05.

Table A3: Average TPR/FPR values for our method and for reference SPC (Guo et al., 2023). Our algorithm has an **automated threshold of 0**. The threshold considered for SPC is 0.5. Results are presented in the format of 'TPR/FPR', where the standard deviation is given in parenthesis, and the values of mean and standard deviation are calculated over 3 runs. Poisoning ratio is 0.05

| Attack | SPC | Ours |
|---|---|---|
| **CIFAR-10** | | |
| Badnet | 0.680 (0.093) / 0.186 (0.008) | 1.000 (0.000) / 0.166 (0.007) |
| Blend | 0.128 (0.013) / 0.186 (0.005) | 0.831 (0.184) / 0.132 (0.022) |
| LabelConsistent | 0.931 (0.010) / 0.190 (0.005) | 0.984 (0.023) / 0.137 (0.049) |
| TUAP | 0.409 (0.163) / 0.182 (0.003) | 0.859 (0.138) / 0.163 (0.044) |
| Trojan | 0.608 (0.028) / 0.172 (0.003) | 0.999 (0.001) / 0.100 (0.001) |
| Wanet | 0.465 (0.028) / 0.185 (0.006) | 0.999 (0.001) / 0.101 (0.001) |

We present the ROC plots **Figure A2** for additional attacks with the poisoning ratio of $\gamma = 0.1$ (not included in **Figure 4**. Additionally, we also provide additional results for poisoning ratio of $\gamma = 0.05$ in **Figure A3**, **Table A3**. Similar to our observations in Section 5.2, our method reaches a near 1 TPR at a much lower FPR value.

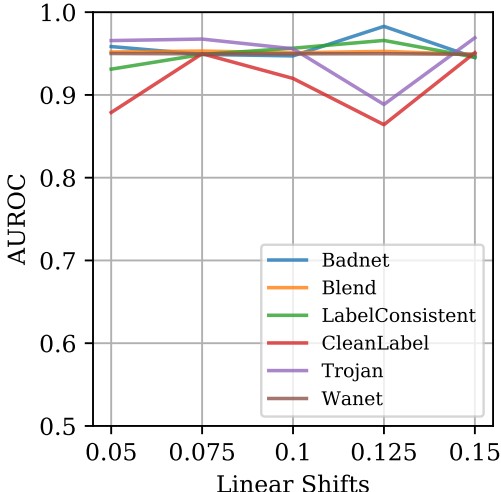

Figure A4: Variation of AUROC for different values of $\tau$ for different backdoor attacks.

## I   RESULTS WITH VIT

We provide additional results using the Badnet attack on various datasets with a vision transformer in **Table A4**. We consider the ViT-Ti/16 (Touvron et al., 2021) pretrained model and finetune it for 50 epochs using Adam optimizer (learning rate = $1e-4$ and weight decay = $5e-4$) and a cosine annealing schedule.

Table A4: Mean AUROC values for our method and for reference SPC (Guo et al., 2023). on CIFAR10, TinyImageNet, and ImageNet200 for the Badnet attack. The best result in each setting is marked in **bold**. All attacks have a poisoning ratio equal to 10%

| Attack | SPC | Ours |
|---|---|---|
| CIFAR-10 | 0.9045 (0.0160) | 0.9798 (0.0025) |
| TinyImageNet | 0.9037 (0.0034) | 0.9986 (0.0008) |
| ImageNet200 | 0.9173 (0.0371) | 0.9982 (0.0006) |

## J   ABLATION STUDY

We have included an ablation study for the hyperparameter $\tau$, which is the linear shift as shown in **Figure A4** of the attached document. We perform this experiment on CIFAR-10 using Resnet-18. We observe that our method achieves an AUROC¿0.9 for most linear shifts across a diverse set of attacks.

## K   ADDITIONAL VISUALIZATIONS

In this section, we present additional figures across a variety of backdoor attacks demonstrating the benefits of the masks (from our bi-level formulation) and algorithm as mentioned in Section 5.2. As shown in **Figure A5**, our optimized masks can in fact, focus on the triggers. For non-patch based triggers like Blend (**Figure A5**-row 2) and Wanet (**Figure A5**-row 6), we hypothesize that our algorithm finds the 'effective part' of the trigger as explained in our formulation of the MSPC loss in Section 4.

We also uncovered various cases of clean images with a high SPC value in Section 4. In **Figure A6**, we present various such examples where our formulation is successful. **Figure A6**-row 1 shows that our masks can avoid the white pixels caused due to the airplane vanishing (Section 4-Scenario 2). As explained in Section 4, our masks can also avoid the spurious correlation (**Figure A6**-row 2). We note that the mask in this case and in **Figure 3** - CIFAR10,row 2 are different because we choose masks

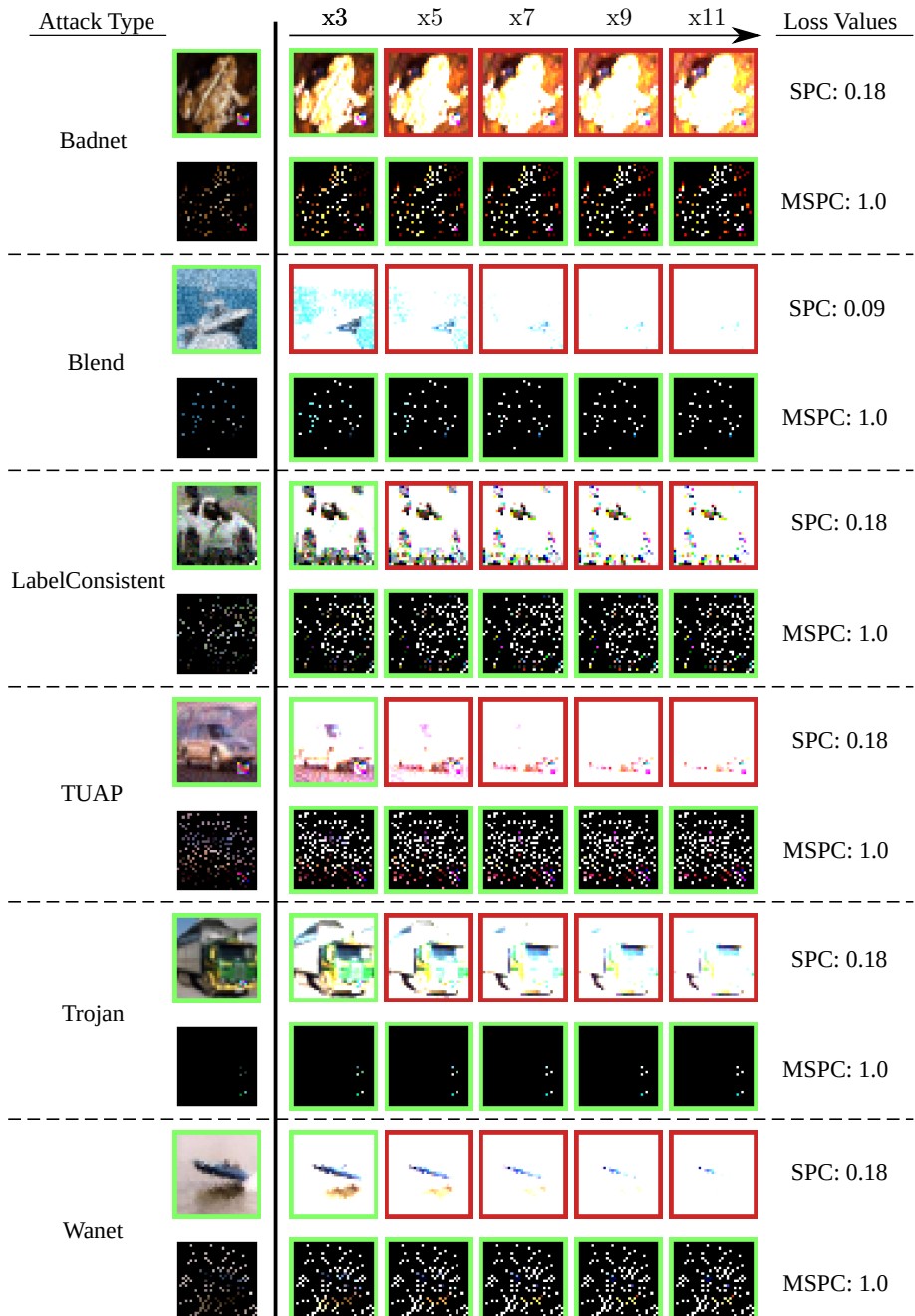

Figure A5: Additional visualizations of **backdoored images** with low SPC loss but high MSPC loss, across different types of backdoor attacks-poisoned training sets. We present examples of the original image and the masked image across scales. The green box indicates that the predicted label is the target label and the red box indicates that the predicted label is different from the target label. We set a threshold of 0.08 for masks for better visualisation.

learnt from different runs. Lastly, our masks can also avoid the generalizing features that stays despite high scales (Section 4-Scenario 1) as seen in **Figure A6**-row 4. Thus, our proposed formulation is successful in improved performance and automatic identification of backdoor datapoints, thus satisfying **P1** and **P2** (Section 1).

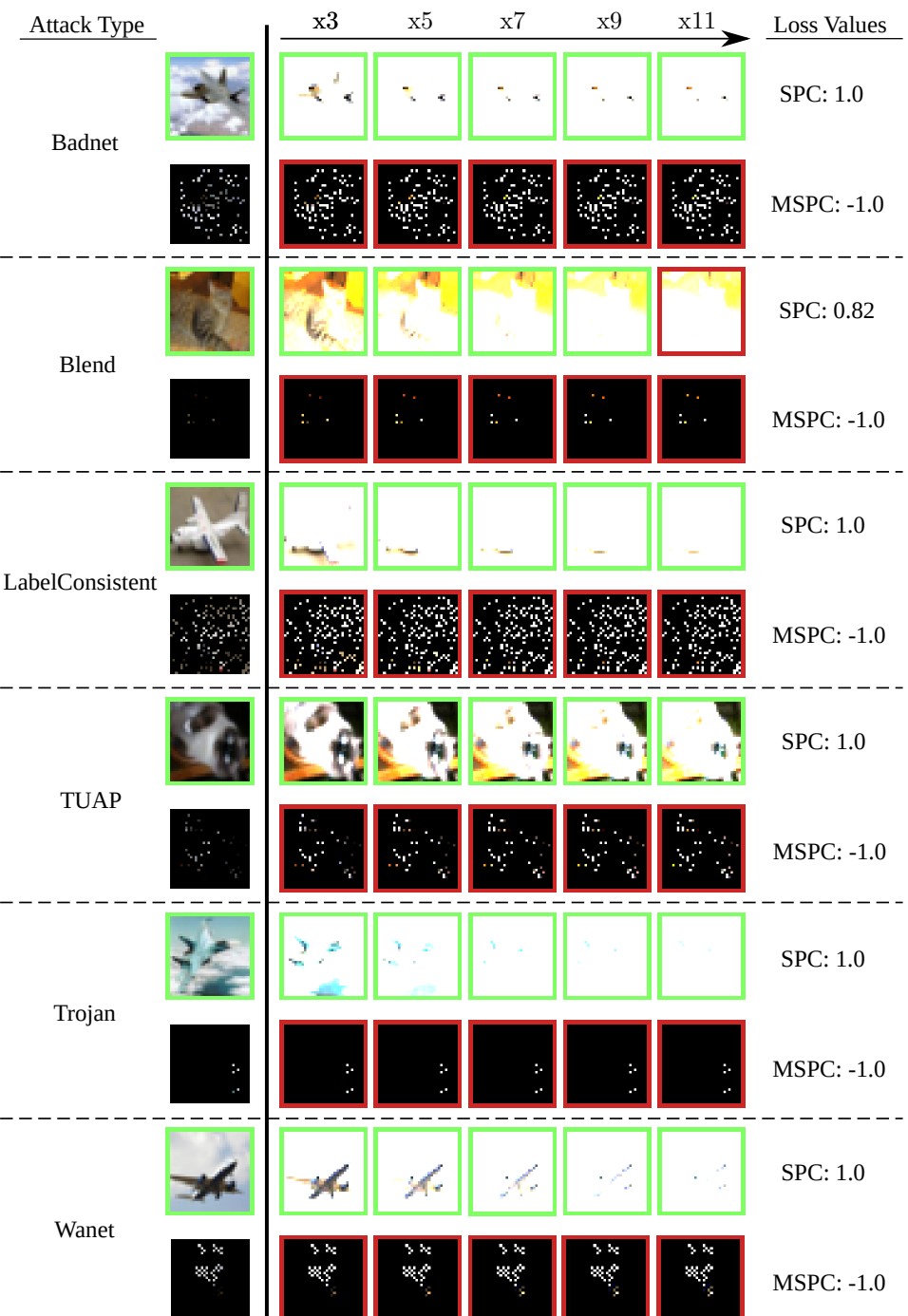

Figure A6: Additional visualizations of **clean images** with high SPC loss but low MSPC loss, across different types of backdoor attacks-poisoned training sets. We use different masks learned after solving our proposed bi-level formulation of (5) as indicated by the attack names. We present examples of the original image and the masked image across scales. The green box indicates that the predicted label is the same as the true class label of the image and the red box indicates that the predicted label is different from the true class label of the image. *We note* that a high SPC indicates backdoor images, and the clean images are predicted as backdoored samples because the prediction remains consistent across scales as indicated by the green boxes. Our proposed MSPC formulation prevents such a phenomenon through the use of our learned masks as indicated by the red boxes. We set a threshold of 0.08 (0.001 for Wanet) for masks for better visualisation.

