# OpenReview forum: "Backdoor Secrets Unveiled: Identifying Backdoor Data with Optimized Scaled Prediction Consistency"
_ICLR.cc/2024/Conference — ICLR 2024 poster_

### Official Review · Reviewer_nd7k · 2023-10-21

**Soundness:** 3 good
**Presentation:** 2 fair
**Contribution:** 2 fair
**Rating:** 6
**Confidence:** 4

**Summary:**

MSPCThis paper proposed a backdoor data detection method called Mask-aware SPC (MSPC). It is inspired by existing work scaled prediction consistency (SPC). Based on the observation that SPC is not robust to the extreme pixel values (near 0 and 1), the proposed method uses a mask and hyperparameter to dynamically adjust the range. Based on the mask, MSPC proposes a bi-level optimization that can detect backdoor samples without the need to specify a threshold. The experiment results show the proposed method is effective in detecting several existing backdoor attacks.

**Strengths:**

- The motivation for adding masks and the linear shift $\tau$ is well explained. It is an interesting observation for applying SPC on images.
- Experiments are comprehensive and include most of the existing attacks; results demonstrated the proposed MSPC is effective in detecting them.

**Weaknesses:**

- For condition P2: Free of Detection Threshold, MSPC is an optimized threshold selection method. The "Free" detection threshold seems ill-defined for the backdoor data detection method, which is essentially a binary classification. Essentially, MSPC still needs to use loss value as the score, except the threshold is optimized through Eq 5.
- After relaxing the mask to continue values, Eq.4 is very similar to CD (Huang et al., 2023). It seems the difference is to replace the absolute difference with the KL divergence. It would be great to include the CD in the experiments for comparison, as well as recent works Meta-SIFT (Zeng et al., 2022) and ASSET (Pan et al., 2023).
- The motivations based on insight 1/2 are constrained by the input bounded from 0 to 1. What happens to SPC if the input is not constrained to 0 to 1 or the input is normalized?
- It has been observed in existing works such as SPECTRE (Hayase et al., 2021) that the detection method is sensitive towards the poisoning rate. It would be more comprehensive to include experiments with lower poisoning rates.

**Questions:**

- Is it possible to incorporate Eq 5 with other detection methods, such as STRIP or ABL?
- What if there are no backdoor samples? What is the FPR in this situation?

---

> ### Author Response · Authors · 2023-11-21
>
> We sincerely thank you for your valuable time and comments. We will alleviate your concerns in this rebuttal as follows.
>
> > **The "Free" detection threshold seems ill-defined for the backdoor data detection method, which is essentially a binary classification.**
>
>
> We'd like to highlight that the baseline methods propose determining this threshold either by utilizing a statistic from a subset of clean data distribution or by employing knowledge of the poisoning ratio or through heuristic methods.
>
> For example, methods like SPECTRE and Spectral Signature assume the knowledge of some upper bound of the poisoning ratio. On the other hand, CD suggests calculating the threshold from the mean and standard deviation of a subset of clean samples. We provide the various existing thresholding methods in Table A1 in the Appendix.
>
> Thus, we think that a backdoor data detection method (which assign scores to training data samples) cannot be trivial unsupervised binary clustering. For example, the scores (which are scalar values) may have multimodal distributions.
>
> > **Essentially, MSPC still needs to use loss value as the score, except the threshold is optimized through Eq 5.**
>
> We would like to point out that the MSPC loss values are used as the score **only after the optimization of Eq. 5 is over**. This is in general not possible if the mask $\mathbf m$ is not optimized.
>
> Moreover, the threshold is **not selected by users** - it is fixed at 0 (by definition) as given by Eq. 2. However, that threshold will not work if we do not have a good mask, which leads to our proposed bilevel formulation of Eq. 5.
>
>
> > **After relaxing the mask to continue values, Eq.4 is very similar to CD (Huang et al., 2023). It seems the difference is to replace the absolute difference with the KL divergence.**
>
>
> We strongly disagree with this statement.
>
> Firstly, the mask in (Huang et al., 2023) represents the minimal mask that is responsible for the same prediction before and after applying the mask, while the mask in Eq.4 is responsible for maintaining the scaled prediction consistency signature.
>
> Notably, Eq.4 includes the variable $\mathbf w$. Based on the fact that the scaled invariance property **is only true for backdoor samples**, Eq. 4 will work if we only consider the backdoor samples. This is the reason why $\mathbf w$ is an important part of the equation and the bilevel formulation of Eq.5 is a natural consequence of moving towards finding that optimal $\mathbf w$.
>
>
>
> > **It would be great to include the CD in the experiments for comparison, as well as recent works Meta-SIFT (Zeng et al., 2022) and ASSET (Pan et al., 2023).**
>
>
> We include results with CD in the general response.
> However, we would like to emphasize that in light of the practical conditions (Section 1 and General response), these are not comparable.
>
> We have highlighted the disadvantage of Meta-SIFT in this problem in Section 5.2 and Figure 5.
>
> It should be noted that Meta-SIFT is developed to identify a small set of **clean samples from the dataset** (as explained in Section 1). To identify backdoor samples accurately, Meta-SIFT needs to identify all clean samples accurately. However, as shown in Fig. 5, it fails to do so (% of Clean Samples Selected = 100) because of the high NCR value.
>
> We should also note that both Meta-SIFT and ASSET violate the practical conditions.
>
>
>
> > **What happens to SPC if the input is not constrained to 0 to 1 or the input is normalized?**
>
> We provide the AUROCs obtained with SPC if the input is not constrained to 0 and 1.
>
> | Badnet         | Blend     |
> | -------------- | --------- |
> | 0.5028 (0.004) | 0.5 (0.0) |
>
> SPC completely fails in such a scenario.
>
>
> > **lower poisoning rates.**
>
> 1. We have performed experiments with low poisoning rates ($\gamma = 0.05$). We have provided these additional results in Appendix G (Fig A3, Table A3).
> 2. Moreover, we point to the Adaptive Blend attack (Table 1, row 8). This attack was specifically designed to work with an extremely low poisoning rate ($\gamma = 0.003$). Our method achieves an AUROC of 0.9046, where all other baselines fail.

---

> > ### Comment · Reviewer_nd7k · 2023-11-22
> > **Response to the authors**
> >
> > Thanks to the authors for the detailed response. The authors addressed my concerns and questions. I have read all reviews from other reviewers and the author's response. I agree with reviewer wHYC that automatic filtering is a good contribution.
> >
> > The proposed MSPC incorporates the idea of the mask used in CD with SPC loss (e.g. replace the same output to scaled prediction consistency signature). This seems to limit its novelty. However, considering the contribution of automatic filtering and solving the limitations of pixel scales in SPC, the reviewer believes the merit outweighs the weaknesses. As a result, I will increase my score to 6.

---

> ### Author Response · Authors · 2023-11-22
> **Thank you.**
>
> Dear Reviewer nd7k,
>
> Thank you very much for the careful review and the valuable comments. It is our great pleasure to learn that your concerns have been addressed and the rating is increased. We will try our best to further improve our paper based on your suggestion.
>
> Thank you very much,
>
> Authors

---

### Official Review · Reviewer_wHYC · 2023-10-31

**Soundness:** 2 fair
**Presentation:** 2 fair
**Contribution:** 2 fair
**Rating:** 6
**Confidence:** 4

**Summary:**

This paper proposes an optimization framework to improve an existing backdoor sample detecting method Scaled Prediction Consistency (SPC). It identifies two limitations of SPC, in terms of the unusual SPC values of backdoor vs. clean samples. It addresses the limitations of SPC by introducing a pre-shift and a learnable mask into SPC, and proposes to use a bilevel optimization framework to first find a small mask and then only scale up the masked region of the image. The new loss function is named Mask-aware SPC (MSPC).  The way to find the minimal mask is similar to the Cognitive Distillation method introduced in (Huang et al. 2023), which should be able to accurately locate the backdoor trigger position. With the learned mask and scaling factor $n$, the authors further propose to automatically identify the backdoored samples from a training dataset with the help of binary variable $w_i$ ($w=1$ for backdoored sample whilst $w=0$  for clean sample). Eventually, if the MSPC loss is >0, then $w_i$ should be 1 to minimize the overall loss $(1-w_i)\cdot L_{MSPC}$. I.e., counting samples with non-negative MSPC loss or $w_i=1$ yields the final backdoor samples. Experiments with 8 backdoor attacks on three datasets demonstrate the effectiveness of the proposed method.

**Strengths:**

1. The proposed method addresses the limitations of an existing work SPC;

2. The experimental results are promising;

3. The method can automatically differentiate backdoor from clean samples via the bilevel optimization framework.

**Weaknesses:**

1. The paper is poorly written, it takes the reviewer to read many times to get the core idea. The relationship to existing works SPC and Cognitive Distillation (CD) should be accurately summarized and discussed. The fundamental/technical difference should be clearly explained.

2. At the beginning of Section 4, it explains why SPC fails the two cases. Yet this was not systematically or quantitatively analyzed. Those are just conjectures.

3. The key technical novelty is the introduction of a learnable mask into the MSPC loss, however, the technique is very similar to an existing backdoor detection method proposed by (Huang et al. 2023).

4. The automatic filtering variable $w_i$ seems unnecessary, as at the end of Section 4, the authors stated that "backdoor samples will simply be the ones with MSPC loss greater than 0".

5. The two proposed practical conditions: 1) free of clean data; and 2) free of detection threshold, look both ok to me. Many defense works assume the availability of a small subset of clean samples, which is quite practical in real-world scenarios. The reviewer understands it is nice to satisfy both conditions but does not think this makes the proposed method fundamentally superior to other detection methods. For condition 2, if all training samples with MSPC loss greater than 0 should be removed, the detector will remove many clean samples when the dataset is extremely clean or dirty (extremely low/high poisoning rates).

6. Strong adaptive attacks should know the mask, it then can adapt itself to have low MSPC loss with multiple surrogate models rather than one, in case to overfit the current model as it did in the "Resistance against Potential Adaptive Attacks." experiments. In other words, generating strong poisons should also be done in a bilevel manner.

7. In Table 1, the detection performance shown on CIFAR-10 is worse than that reported in the Cognitive Distillation (CD) paper (Huang et al. 2023) (their AUC is above 90%), and the results on test (and poisoned) samples should also be reported.   This means the use of CD in the proposed way actually hurts the detection performance, which may be caused by the automatic search process with $w_i$ and the SPC loss.

**Questions:**

See weaknesses above.

**Details Of Ethics Concerns:**

No Ethics Concerns.

---

> ### Author Response · Authors · 2023-11-21
>
> We sincerely thank you for your valuable time and comments. We will alleviate your concerns in this rebuttal as follows.
>
> > **The paper is poorly written, it takes the reviewer to read many times to get the core idea.**
>
> Thank you for this feedback. We respectfully disagree. We have made a substantial effort to improve the presentation of the paper, including clarifying practical assumptions, providing more illustrative examples, and illustrating the rationals behind our proposal.  However, we are always eager to learn and would greatly appreciate any specific suggestions you might have on how we can enhance our writing.
>
> Additionally, we would like to gently mention that Reviewer YD42 and Reviewer V1zd have commended our paper for being well-written and easy to follow.
>
>
> > **key technical novelty is the introduction of a learnable mask into the MSPC loss**
>
> We would like to point out that the **we introduce the MSPC loss**. So the key technical novelty are:
> 1. Introduction of the MSPC loss (Eq 2)
> 2. Development of the bilevel formulation (Eq 5), which finds the optimal mask and automatically identifies backdoor datapoints.
> **We also highlight our overall contributions at the end of Section 1.**
>
> > **the technique is very similar to an existing backdoor detection method proposed by (Huang et al. 2023).**
>
> We would strongly disagree with this statement.
>
> We would like to point out that the only similarity with (Huang et al. 2023) is that we also penalize the l1 norm of the mask.
>
> As mentioned before, our key technical contributions are the introduction of the novel MSPC loss and the novel bilevel formulation. This also helps us in satisfying P1 and P2, whereas (Huang et al. 2023) assumes the presence of clean data for identifying backdoor datapoints (thus violating P1).
>
>
> > **The automatic filtering variable  seems unnecessary, as at the end of Section 4, the authors stated that "backdoor samples will simply be the ones with MSPC loss greater than 0".**
>
> We would emphasize that the statement "backdoor samples will simply be the ones with MSPC loss greater than 0" is only true **after** the optimization of Eq 5 is complete.
>
> We introduce the automatic filtering problem as a bilevel problem in Eq. 3, which contains the variable $\mathbf w$. Without this variable, it would be impossible to formulate our problem as a bilevel optimization.
>
> > **The reviewer understands it is nice to satisfy both conditions but does not think this makes the proposed method fundamentally superior to other detection methods.**
>
> We elucidate the importance of the practical conditions in the General response.
> Given the existence of scenarios where it can be difficult / impossible to collect clean data, we believe that algorithms satisfying P1 and P2 will be favourable to practitioners.
>
>
> > **In Table 1, the detection performance shown on CIFAR-10 is worse than that reported in the Cognitive Distillation (CD) paper (Huang et al. 2023) (their AUC is above 90%), and the results on test (and poisoned) samples should also be reported.**
>
>
> 1. We would like to point out that our AUROC for CIFAR-10 is also more than 90% for all attacks other than DFST.
> 2. We aim to identify backdoor datapoints from the *training set*. If identified correctly, the user can either remove those backdoor data or unlearn them (as mentioned in 'Model Retraining Effect.' in Section 5.2).
>
>
> > **This means the use of CD in the proposed way actually hurts the detection performance, which may be caused by the automatic search process with  and the SPC loss.**
>
> Firstly, we would like to emphasize that we **do not** use CD. Our contributions include introduction of the MSPC loss (Eq 2) and development of the bilevel formulation (Eq 5).
>
> We added experiments using CD in our general response and compared its performance with ours. We would like to highlight the following points:
> 1. CD performs well in cifar10. However, **CD fails completely for the recently introduced attack Adaptive Blend**. However, our method maintains an AUROC > 90% for this attack.
> 2. Our method performs **better than CD in all attacks for both TinyImagenet and Imagenet200**.
> 3. Most importantly, CD violates P1 from our practical conditions. Backed by our discussion of practical conditions (Section 1, Table 1 and General Response), we emphasize the fact that we cannot compare CD and our method fairly because CD uses clean data to find the threshold but ours do not.

---

> > ### Comment · Reviewer_wHYC · 2023-11-22
> > **Response to the authors**
> >
> > Thanks to the authors for the detailed response. Although I realized certain misunderstandings of the proposed idea with the clarifications, the authors are still encouraged to make the idea clearer at the beginning of the paper. E.g., without sufficient reasoning, it is hard to accept the two conditions but rather to believe they are tricks to avoid comparison to existing works. I.e., automatic filtering can be made into an independent process on top of any detection methods. But I agree with the authors that having an automatic detection method free of hyperparameters can be a good contribution to the community. Considering the MSPC loss is prosed in this work, I am ok to increase my rating.

---

> ### Author Response · Authors · 2023-11-22
> **Thank you**
>
> Dear Reviewer wHYC,
>
> Thank you very much for the careful review and the valuable comments. It is our great pleasure to learn that your concerns have been addressed and the rating is increased.  We are happy to know that you consider our proposed automated detection is a good contribution. We will try our best to further improve our paper based on your suggestion.
>
> Thank you very much,
>
> Authors

---

### Official Review · Reviewer_V1zd · 2023-10-31

**Soundness:** 3 good
**Presentation:** 3 good
**Contribution:** 2 fair
**Rating:** 6
**Confidence:** 4

**Summary:**

The paper presents a method for backdoor detection that identifies poisoned samples based on prediction invariance after scaling. The method designed a new bi-level optimization-based approach to improve the performance of the existing SPC method.

**Strengths:**

- The manuscript is well-organized and straightforward, facilitating easy comprehension.
- The empirical results are compelling and substantiate the paper's claims effectively.
- I had previously reviewed this work for an earlier conference. Given the improvements the authors have made—specifically, the expansion of datasets and attack baselines—I am inclined to give it a “marginally above the acceptance threshold”.

**Weaknesses:**

- Despite some improvements in this submission, my primary concern remains: the core idea behind the proposed method is still closely aligned with the existing Scaled Prediction Consistency (SPC) approach, limiting the paper's technical novelty and contribution.
- Extending the experiments to include more diverse model architectures, such as attention-based ViT, would bolster the robustness of the findings. Currently, only one model architecture (ResNet-18) is used for experimental evaluation.

**Questions:**

No further questions at this time.

---

> ### Author Response · Authors · 2023-11-21
>
> We sincerely thank you for your valuable time and positive feedback. We will alleviate your concerns in this rebuttal as follows.
>
> > **closely aligned with the existing Scaled Prediction Consistency (SPC) approach**
>
> We would like to argue that our work leverages this signature and provides a non-trivial algorithm to identify backdoor data samples without any additional assumptions as discussed in Section 1 , Table 1, General Response.
>
> 1. Our proposed algorithm shows massive amounts of improvements over SPC across a variety of attacks as shown in Table 1.
> 2. We argue that our work propose in a novel algorithm that can identify backdoor data obeying the practical conditions, while SPC is incapable of doing so.
>
>
>
> > **Extending the experiments to include more diverse model architectures, such as attention-based ViT**
>
>
> We have included results with ViT-Ti/16 [1] in the general response. As shown in the results, our method achieves similar AUROC values (sometimes better) when compared to Resnet18. This demonstrates the robustness of our method across different architectures.
>
>
> 1. Touvron, Hugo, et al. "Training data-efficient image transformers & distillation through attention." International conference on machine learning. PMLR, 2021.

---

> > ### Comment · Reviewer_V1zd · 2023-11-22
> > **Reply to authors**
> >
> > Thank the authors for their response. Although I have doubts about the novelty and technical contribution of this work, considering the improvement in effectiveness, I am slightly inclined to accept this paper.

---

### Official Review · Reviewer_YD42 · 2023-10-31

**Soundness:** 2 fair
**Presentation:** 2 fair
**Contribution:** 2 fair
**Rating:** 6
**Confidence:** 5

**Summary:**

This study focuses on automatically detecting backdoor data in poisoned machine learning datasets, without requiring clean data or predefined thresholds. It leverages the scaled prediction consistency (SPC) technique, introducing a unique SPC-based loss function for precise identification. This research addresses limitations in the traditional SPC method and develops a bi-level optimization approach for accurate backdoor data detection. The proposed method is evaluated on several datasets.

**Strengths:**

1. The problem of backdoor sample identification is of sufficient interests for the community.
2. The paper is well-written and easy to follow.

**Weaknesses:**

1. Regarding the AUROC performance, where a comprehensive threshold iteration is conducted, the proposed method exhibits only marginal improvements over the STRIP method. Notably, in the case of CIFAR-10, the proposed method significantly outperforms the STRIP method solely under the AdapBlend (γ = 0.3%) condition. Surprisingly, in the context of Tiny ImageNet, the STRIP method even surpasses the proposed method.
2. Could you provide a runtime analysis of the algorithms employed to solve the bi-level optimization problem? Additionally, it would be valuable to understand the optimization process for the discrete variable w?
3. While it is acknowledged that running the STRIP method on ImageNet 200 presents time complexity challenges, I would like to point out that this method does not appear to encounter runtime issues comparable to your proposed methods, which require solving a discrete bi-level optimization problem. In the STRIP method, the procedure only involves *superimposing two images and forwarding them to the backdoored model to obtain the outputs*.

In light of the aforementioned observations, it appears that the proposed method does not introduce significant advantages, either in terms of computational complexity or performance enhancement, when compared to the STRIP method. Consequently, I recommend rejection.

**Questions:**

Please see my comments above.

---

> ### Author Response · Authors · 2023-11-21
>
> We sincerely thank you for your valuable time and comments. We will alleviate your concerns in this rebuttal as follows.
>
> > **Notably, in the case of CIFAR-10, the proposed method significantly outperforms the STRIP method solely under the AdapBlend (γ = 0.3%) condition. Surprisingly, in the context of Tiny ImageNet, the STRIP method even surpasses the proposed method.**
>
> We would like to point out that this is in fact **incorrect**.
> 1. Our method surpasses STRIP in both Wanet and AdapBlend for CIFAR-10. Notably, STRIP completely fails in AdapBlend (AUROC = 0.1378).
> 2. In Tinyimagenet, our method is better than STRIP for both Badnet and Wanet.
> 3. For Imagenet200, our method outperforms STRIP on average (as given in the general response).  STRIP's performance considerably suffers for Badnet attack on Imagenet when compared to our method.
>
> > **Could you provide a runtime analysis of the algorithms employed to solve the bi-level optimization problem?**
>
> We provide runtime (in mins) for our algorithm for different datasets.
>
> We would like to point out that our runtime is dependant on the number of epochs we perform for solving the inner optimization (Epoch_in) and the number of overall epochs (Epoch_out) for solving the bilevel optimization.
>
> We present the time and AUROC values for Badnet attack for different datasets and different configurations of numbers of epochs. We find that our method maintains high AUROC values across various configurations of epoch numbers.
>
>
>
> | Dataset      | Epoch_in | Epoch_out | AUROC           | Time (mins) |
> | ------------ | -------- | --------- | --------------- | ----------- |
> | TinyImagenet | 5        | 4         | 0.9983 (0.0004) | 147         |
> | TinyImagenet | 2        | 2         | 0.9998 (0.0006) | 37.2        |
> | Imagenet200  | 5        | 4         | 0.9980 (0.0004) | 164.71      |
> | Imagenet200  | 2        | 2         | 0.9977 (0.0005) | 40.9        |
> | cifar10      | 10       | 2         | 0.9514 (0.0043) | 35.64       |
> | cifar10      | 2        | 2         | 0.9562 (0.006)  | 4.8         |
>
>
>
> > **STRIP runtime**
>
> We have included new results for STRIP with the Imagenet200 dataset (given in the general response). For comparison, this takes about 70 minutes.
>
>
> > **proposed method does not introduce significant advantages, either in terms of computational complexity or performance enhancement, when compared to the STRIP method.**
>
> We strongly disagree with this statement. We introduce two key innovations:
> 1. Introduction of the MSPC loss (Eq 2)
> 2. Development of the bilevel formulation (Eq 5), which finds the optimal mask and automatically identifies backdoor datapoints.
>
> **We also highlight our overall contributions at the end of Section 1.**
>
> We are not claiming advantages of computation complexity nor massive performance gains. As explained in Section 1 , Table 1 and in our general response, our method satisfies the practical constraints which other baseline methods do not. Even while doing so, *our  approach often outperforms or performs at par with baseline methods (which do not satisfy the practical constraints)*.

---

> > ### Comment · Reviewer_YD42 · 2023-11-22
> > **Thanks for the authors' rebuttal; further questions**
> >
> > I would like to thank for the authors' respones for my comments as well as additional experiments.
> >
> > In simple terms, I think the two assumptions the literature relies on are okay, as pointed out by others too. So what I'm really looking at is how well it performs and the computational side of things. When it comes to performance and computational aspects, I have doubts about some of the claims made by the authors—they're not very exact, especially without proper statistical backing.
> >
> >
> > - I would like to first respond certain points in response for the authors' reply:
> >
> >
> > > 'In Tinyimagenet, our method is better than STRIP for both Badnet and Wanet.'
> >
> > This may not hold true. Looking at the means and stds reported for Badnet and Wanet in Table 2, your statement might not have statistical significance. To illustrate, if you conduct independent experiments ten times and run a t-test with a set significance level of 0.05, you won't be able to reject the null hypothesis. This result indicates that there's no statistically significant difference between the AUROC means of your method and STRIP, making your assertion invalid.
> >
> > Could you kindly share the number of independent runs for further clarification?
> >
> >
> > > `Our method surpasses STRIP in both Wanet and AdapBlend for CIFAR-10. Notably, STRIP completely fails in AdapBlend (AUROC = 0.1378).'
> >
> > While it's true that your approach performs better than STRIP Wanet and AdapBlend for CIFAR-10, your Table 2 indicates that the STRIP method surpasses your approach in five other attacks (Blend, LC, TUAP, Trojan, and DFST). Although your method currently shows higher averaged scores than STRIP, this is primarily due to AdapBlend's failure case. When considering a broader range of attacks, it's not necessarily guaranteed that your method will outperform STRIP in terms of overall performance.
> >
> > Moreover, in terms of failure cases, your method can also falter under the DFST attack. Hence, at least from an effectiveness standpoint, your method doesn't seem superior to STRIP. I acknowledge that your method is built on less stringent assumptions, but for a more accurate interpretation of your results, it's crucial to articulate your claims precisely and rigorously.
> >
> >
> > - I have additional inquiries. Could you provide the training loss curve for your optimization procedure? I am concerned that your choice of both inner and outer rounds is relatively small, which may lead to insufficient convergence of the training loss. Consequently, it seems that your method may still face a substantial computational burden.
> >
> > If the authors can address my concerns above, I will raise my score.

---

> ### Author Response · Authors · 2023-11-23
>
> We thank the reviewer for their reply and valuable comments. We hope to alleviate the additional concerns in this rebuttal.
>
>
> > **Looking at the means and stds reported for Badnet and Wanet in Table 2, your statement might not have statistical significance.**
> >
> > **Could you kindly share the number of independent runs for further clarification?**
>
> Thank you very much for the suggestion. Currently, our results are obtained over 3 independent runs. Encouraged by the comments, we conducted 2 more runs and performed a Welch's one-tailed t-test between the AUROC values of Badnet and Wanet. Our results have a significance level of 0.08 and 0.06 (<0.1) for these attacks in Tinyimagenet, respectively.
>
> We will confirm this significance result with 10 independent runs later and can update it upon the reviewer's request.
>
>
> > **Hence, at least from an effectiveness standpoint, your method doesn't seem superior to STRIP. I acknowledge that your method is built on less stringent assumptions, but for a more accurate interpretation of your results, it's crucial to articulate your claims precisely and rigorously.**
>
>
> Thank you for this suggestion. We want to clarify that we did **not** intend to assert any advantages in terms of computational efficiency or promise absolute massive performance gains. As the reviewer pointed out, "I acknowledge that your method is built on less stringent assumptions," and as we explained in Section 1, our achieved performance, which is on par with or potentially outperforming baseline methods, is contingent upon practical conditions. It's important to note that the baseline methods may not necessarily meet these same conditions. In this context, it holds particular significance for us because our method, which relies on fewer assumptions, manages to achieve highly competitive performance.
>
>
>
> **About STRIP**:
> Please allow us to make further clarifications on this matter.
> The AUROC (Area Under the Receiver Operating Characteristic) is determined by calculating the area under the curve of the True Positive Rate (TPR) and False Positive Rate (FPR) across various detection thresholds. Consequently, when employing a backdoor identification method, it is not necessary to find a precise detection threshold for AUROC computation
>
>
> The STRIP work suggests to find this detection boundary **using the mean and variance of the entropy distribution of clean samples**. In contrast, our method can automatically detect the backdoor samples without the necessity of clean samples.
>
> Therefore, the computation of AUROC values for STRIP does not require detection boundary computation. Consequently, in scenarios where clean samples are absent, this limitation does not manifest in the AUROC values, as they are computed by directly utilizing the entropy values of all samples.
>
> Considering the fact that AUROC offers a limited representation of the overall problem, we posit that relying solely on this performance metric may not be a comprehensive measure of the success of our method. We appreciate your consideration of these points.
>
>
>
>
>
>
>
> > **Could you provide the training loss curve for your optimization procedure? I am concerned that your choice of both inner and outer rounds is relatively small, which may lead to insufficient convergence of the training loss.**
>
> Thank you for raising this question. We consider 2 epochs of outer optimization and 2 epochs of inner optimization.
>
> We present the masking loss convergence curve for CIFAR10 in this [Figure](https://ibb.co/QNVNTH6) and in this [Figure](https://ibb.co/QPNYZV2) for TinyImagenet. For CIFAR10, each epoch contains 50 iterations, resulting in a total of 200 iterations, while for TinyImagenet, there are a total of 400 iterations because of 100 iterations per epoch.
> We will present similar loss curves for Imagenet200 soon, if needed.

---

### Author Response · Authors · 2023-11-21
**General Response 1 : Importance of Practical Conditions**

# Importance of the practical conditions

We would like to emphasize the importance of the practical conditions that we introduce in our paper. This also shows that previous methods are infact not comparable. Still, we have included them for reference in Table 1.

**Violation of Practical Condition 2 (P2)**:

This requires the user to set some threshold to identify backdoor datapoints. Such a threshold depends on various factors like the poisoning ratio, the backdoor attack type and the dataset being used.

For example, methods like SPECTRE and Spectral Signature assume the knowledge of some upper bound of the poisoning ratio. On the other hand, CD suggests calculating the threshold from the mean and standard deviation of a subset of clean samples. We provide the various existing thresholding methods in Table A1 in the Appendix.

Thus, without some form of prior knowledge like the poisoning ratio (*which is seldom the case*), it is impossible for an user to determine such a threshold value.

**Violation of Practical Condition 1 (P1)**:

We provide several examples to understand the importance of the clean data assumption:

* Firstly, we will provide an example **where it may be very hard / impossible to obtain clean data**.
We consider the case of prediction of the presence of Autism Spectrum Disorder (ASD) in children. Recent studies [1] collect home videos of children (submitted by parents) via crowdsourcing. Parents are required to take home videos of their child and upload them to a portal. Such crowdsourcing methods maybe highly susceptible to backdoor attacks.

In such cases, it may be very hard to collect clean data because that would require monitoring of the video capturing and uploading. This can be difficult or impossible because of various reasons (eg. change in behaviour of children, inavailability of consent to monitoring etc.)

* In some cases, organizations (which are clients to various ML companies) may be **reluctant to provide / collect clean data** for their application. For example, in a recent industry survey [2], companies included backdoor attacks as one of their security concerns. We directly quote this study:
> organizations seem to push the security responsibility upstream to service providers as one of the respondents said, “We use Machine Learning as a Service and expect them to provide these robust algorithms and platforms”



We believe many such scenarios associated with impossibility or reluctance of clean data collection may arise and are only limited to our imagination. This necessitates the development of algorithms without clean data assumption.

Backed by such arguments, we emphasize that **it is difficult to directly compare our approach with baselines in a fair manner** as the former satisfies the practical conditions and the latter violates P1 or P2 or both.



**References**


1. Tariq Q, Daniels J, Schwartz JN, Washington P, Kalantarian H, Wall DP. Mobile detection of autism through machine learning on home video: A development and prospective validation study. PLoS Med. 2018 Nov 27;15(11):e1002705. doi: 10.1371/journal.pmed.1002705. PMID: 30481180; PMCID: PMC6258501.
2. Grosse, Kathrin, et al. "Machine learning security in industry: A quantitative survey." IEEE Transactions on Information Forensics and Security 18 (2023): 1749-1762.

---

### Author Response · Authors · 2023-11-21
**General Response 2: Additional Results**

# STRIP results for Imagenet200

We completed experiments for Imagenet200 using STRIP and include the obtained results.

| Badnet          | Blend           | Average |
| --------------- | --------------- | ------- |
| 0.8681 (0.0419) | 0.9916 (0.0024) | 0.9298  |

Notably, we observe that the performance of STRIP considerably degrades for Badnet attack on Imagenet when compared to our method.



# Cognitive Distillation comparison results

We provide AUROC values for different attacks with different datasets (as in Table 1) for Cognitive Distillation.



| Dataset      | Attack                        | AUROC           |
| ------------ | ----------------------------- | --------------- |
| CIFAR-10     | Badnet                        | 0.9998 (0.0000) |
| CIFAR-10     | Blend                         | 0.9490 (0.0122) |
| CIFAR-10     | LabelConsistent               | 0.9980 (0.0013) |
| CIFAR-10     | TUAP                          | 0.9897 (0.0018) |
| CIFAR-10     | Trojan                        | 0.9974 (0.0007) |
| CIFAR-10     | Wanet                         | 0.9892 (0.0024) |
| CIFAR-10     | DFST                          | 0.8693 (0.0388) |
| CIFAR-10     | AdapBlend ($\gamma = 0.3 \%$) | 0.2139 (0.0232) |
| TinyImagenet | Badnet                        | 0.9629 (0.0017) |
| TinyImagenet | Blend                         | 0.9884 (0.0004) |
| TinyImagenet | Wanet                         | 0.9887 (0.0070) |
| Imagenet200 | Badnet                         | 0.5410 (0.0504) |
| Imagenet200 | Blend                        | 0.9013 (0.0219) |


1. CD performs well in cifar10. However, **CD fails completely for the recently introduced attack Adaptive Blend**. However, our method maintains an AUROC > 90% for this attack.
2. Our method performs **better than CD in all attacks for both TinyImagenet and Imagenet200**.
3. Moreover, backed by our discussion of practical conditions, we emphasize the fact that we cannot compare CD and our method fairly because CD uses clean data to find the threshold but ours do not.

# Results with ViT


We provide results for Badnet attack for various dataset with a vision transformer. We consider the ViT-Ti/16 [1] pretrained model and finetune it for 50 epochs using Adam optimizer (learning rate = 1e-4 and weight decay = 5e-4) and a cosine annealing schedule.


| Dataset      | AUROC           |
| ------------ | --------------- |
| CIFAR-10     | 0.9798 (0.0025) |
| TinyImagenet | 0.9986 (0.0008) |
| Imagenet200 | 0.9982 (0.0006) |


Thus, these show the robustness of our method for various architectures including resnets and attention based vision transformers.


[1] Touvron, Hugo, et al. "Training data-efficient image transformers & distillation through attention." International conference on machine learning. PMLR, 2021.

---

### Meta-Review · Area_Chair_ceqf · 2023-12-05

**Metareview:**

This paper studies an interesting problem in backdoor learning: the automatic identification of backdoor data within a poisoned dataset, without the need for additional clean data or manually defining a threshold for backdoor detection. Inspired by a previous work that uses scaled prediction consistency, they propose a new SPC-based loss and solve this data identification process through bilevel optimization. The proposed method can thus identify backdoor data crafted from different methods, and surpasses the baseline defenses in a considerable amount.

After discussion, the reviewers generally agree that the authors have adequately addressed their concerns on the proposed method, and the proposed method, although has limited technical novelty compared to previous approaches, does deliver promising results and contribute to a good defense strategy. Based on these opinions, I thus recommend accepting the paper, and would urge the authors to incorporate the comments and new results in the final version.

**Justification For Why Not Higher Score:**

As pointed out by many reviewers, the proposed approach is more of an extension of the previous SPC method, and also has limited generality.

**Justification For Why Not Lower Score:**

This paper does provide good defense performance under the rather restrictive setting, and performs well against different backdoor injection methods.

---

### Decision · Program_Chairs · 2024-01-16

Accept (poster)